# Hierarchical Resource Management for Mega-LEO Satellite Constellation

**DOI:** 10.3390/s25030902

**Published:** 2025-02-02

**Authors:** Liang Gou, Dongming Bian, Yulei Nie, Gengxin Zhang, Hongwei Zhou, Yulin Shi, Lei Zhang

**Affiliations:** 1School of Communications and Information Engineering, Nanjing University of Posts and Telecommunications, Nanjing 210003, China; 2College of Integrated Circuit Science and Engineering, Nanjing University of Posts and Telecommunications, Nanjing 210023, China; 3Institute of Telecommunication and Navigation Satellite, China Academy of Space Technology, Beijing 100094, China

**Keywords:** mega-LEO satellite constellation system, hierarchical resource management, spectrum cognition, interference avoidance, service area planning

## Abstract

The mega-low Earth orbit (LEO) satellite constellation is pivotal for the future of satellite Internet and 6G networks. In the mega-LEO satellite constellation system (MLSCS), which is the spatial distribution of satellites, global users, and their services, along with the utilization of global spectrum resources, significantly impacts resource allocation and scheduling. This paper addresses the challenge of effectively allocating system resources based on service and resource distribution, particularly in hotspot areas where user demand is concentrated, to enhance resource utilization efficiency. We propose a novel three-layer management architecture designed to implement scheduling strategies and alleviate the processing burden on the terrestrial Network Control Center (NCC), while providing real-time scheduling capabilities to adapt to rapid changes in network topology, resource distribution, and service requirements. The three layers of the resource management architecture—NCC, space base station (SBS), and user terminal (UT)—are discussed in detail, along with the functions and responsibilities of each layer. Additionally, we explore various resource scheduling strategies, approaches, and algorithms, including spectrum cognition, interference coordination, beam scheduling, multi-satellite collaboration, and random access. Simulations demonstrate the effectiveness of the proposed approaches and algorithms, indicating significant improvements in resource management in the MLSCS.

## 1. Introduction

In recent years, low-cost rocket launches and advanced manufacturing technologies have facilitated the deployment of satellite Internet, particularly through low Earth orbit (LEO) satellites. The low orbital altitude of LEO satellites results in minimal space–ground transmission delays, making them well-suited for delay-sensitive services and application scenarios, such as multimedia entertainment, the Internet of Vehicles (IoV), and real-time control. Additionally, LEO satellites enable terminal miniaturization due to low propagation loss, while dense constellation deployment ensures seamless global coverage. Consequently, the MLSCS has garnered significant attention from governments and global industries, leading to extensive deployments in recent years [1].

The Starlink system, developed by Space Exploration Technologies (SpaceX) and OneWeb, serves as a prominent example of satellite Internet deployment. SpaceX initially planned to deploy two sub-constellations, consisting of 4425 satellites at an orbital altitude of 1150 km and 7518 satellites at an orbital altitude of 340 km, to provide V-band and E-band broadband Internet access services for global users. In November 2018, SpaceX adjusted its orbital altitude from 1150 km to 550 km [2] and the latest plan includes launching 42,000 satellites at multiple different orbital altitudes. By 30 October 2024, the total number of Starlink satellites has reached 7213, of which more than 1400 were launched in 2024. According to the ITU milestone requirements, the SpaceX needs to complete the launch of 1193/5963/11,926 satellites in the Starlink Phase I&II by 2025/2028/2030, and the launch of 3000/15,000/30,000 satellites in the Starlink Phase III by 2028/2031/2033. Elon Musk, the CEO of SpaceX, announced on platform X (formerly Twitter) that Starlink has achieved breakeven cash flow on 2 November 2023.

The OneWeb constellation offers users impressive connectivity capabilities, providing up to 400 Mbps downlink and 30 Mbps uplink rates. This high-speed Internet access is designed to support a variety of vertical application scenarios, including aviation, maritime operations, and cellular communications. Different types of terminals can be utilized to meet the specific needs of these sectors, enhancing connectivity and enabling innovative solutions across diverse industries [3,4]. The OneWeb constellation provides users with global seamless coverage and commercial satellite access. As of now, OneWeb has successfully launched 618 of its planned 648 satellites, indicating that the deployment of the first orbital shell of its space network is nearing completion. This achievement is a significant step towards enhancing global connectivity and expanding access to satellite Internet services.

Radio resource management is a key technology in wireless communication systems, significantly impacting the performance of the MLSCS. As the number of system users and services increases, along with the expansion of application scenarios, it becomes essential to optimize radio resource scheduling to enhance satellite efficiency and steadily improve network capacity. Therefore, the architectural design of radio resource management systems in the MLSCS requires functional decoupling and scalability to meet these evolving demands.

Unlike traditional geosynchronous Earth orbit (GEO) satellite communication systems, the coverage area of satellites in the MLSCS changes dynamically. In comparison to constellation satellite communication systems with fixed beams, such as Iridium and OneWeb, some LEO satellite constellation systems exhibit insufficient service beam coverage. This limitation increases the need for on-demand service beam scheduling, known as beam-hopping. Compared to traditional satellite communication systems, the application of beam-hopping can address the imbalanced characteristics of service distribution. Through effective beam scheduling, interference with GEO satellites can be avoided, and on-demand services can be realized, significantly reducing coverage in areas without service and optimizing overall system performance.

Due to the size and platform limitations of LEO satellites, their single-satellite capability are relatively constrained. Hence, multi-satellite collaborative service enhancement is crucial to LEO satellite constellation. This includes multi-satellite collaborative scheduling to compensate for gaps in coverage caused by interference avoidance with GEO satellites, as well as multi-satellite service enhancement to improve service capabilities in hotspot areas. The purpose of radio resource scheduling in the MLSCS is to enable dynamic multi-satellite collaborative scheduling of at SBSs based on global service demands and resource usage. Additionally, multi-satellite radio resource scheduling serves as an effective approach to address the limitations of single-satellite service capacity and achieve global high-speed data service.

To date, there is no global architecture or comprehensive scheme for managing radio resource management in the MLSCS that can effectively schedule various radio resources from space to ground, including spectrum, beams, slots and power. Currently, the competition for spectra among different systems, satellites and beams, as well as the competition for power, slots, and bands among users, cannot be effectively coordinated. Additionally, the challenges of intra-system and inter-system interference, along with mobility management in a highly dynamic environment, have not been fully addressed.

To enhance radio resource management in the MLSCS, the following objectives are essential: (1) analyze the correlation and influencing factors of radio resource management; (2) establish a uniform and comprehensive radio resource management architecture with effective strategies and processing procedures from the NCC to SBSs and user terminals; (3) analyze and arrange the functions and procedures of each component in the system; (4) propose detailed and feasible strategies, approaches, and algorithms for specific functions and procedures; and (5) establish an indicator system to assess the effectiveness and efficiency of the resource scheduling architecture, strategies, approaches, and algorithms.

Despite advancements in satellite communication, comprehensive resource management solutions remain underexplored. To date, there have been no comprehensive achievements in this area; current research primarily focuses on specific functions or modules. This gap motivates us to investigate these issues in this paper. We conduct a review of the existing literature on resource management in satellite communication networks and space information networks, particularly focusing on the MLSCS, and present the following contributions:To design and propose a resource management architecture, strategies, approaches, and algorithms, the influencing factors for resource scheduling and usage are presented first. This understanding serves as the basis for subsequent proposals aimed at enhancing resource management in satellite communication networks.We study and design a hierarchical resource management architecture (at the NCC, SBSs, and user terminals, respectively) according to resource distribution and application requirements at different levels, so that resource scheduling is more flexible, efficient, and real-time compared with unified implementation in the whole network.As part of our research, the functions and procedures of each component will be analyzed and depicted in detail.In this paper, we present the approaches and algorithms associated with the main components, functions, and procedures of our proposed system. These methodologies aim to enhance efficiency and effectiveness in resource management. For spectrum planning at the NCC, spectrum data mining methods based on causal regression, LSTM prediction, and anomaly detection are presented, and spectrum planning based on game theory, anomaly algorithms, machine learning, and knowledge graphs is summarized. Service area planning needs to comprehensively consider multiple parameters, which can be obtained by multiple algorithms, such as satellite coverage area calculation, interference avoidance calculation, and residual time calculation, etc. Then, a heuristic beam scheduling algorithm is studied for beam-hopping satellite systems.In this study, we provide an assessment of effectiveness and efficiency, along with the primary performance indicators. These metrics are essential for evaluating the overall performance of our proposed system. We conduct some simulations to validate the effectiveness and efficiency of our architecture, approaches, and algorithms.

The organization of this paper is presented as follows: In Section 2, we discuss related works on resource management in satellite communication systems and SINs. Section 3 examines the correlation and influencing factors of resource management in the MLSCS. Section 4 proposes a three-layer architecture for implementing system resource management. Detailed functions and procedures are described in Section 5. In Section 6, we depict the approaches and algorithms for each component and procedure. Section 7 presents performance assessments and indicators. Finally, Section 8 concludes the paper and outlines future work.

## 2. Related Works

Resources (slots, beams, spectrum, power, etc.) play an important role in communication networks. Resources must be used at the right time and in the right places to fully realize the potential of communication networks, while also minimizing resource consumption. As a result, rational and optimized resource scheduling has become crucial for satellite communication networks, with several significant outcomes presented in recent years.

In [5], a multi-layer LEO-MSS (mobile satellite system) architecture has been proposed, along with research on resource optimization to enhance network capacity. A novel hierarchical resource management framework and a joint dynamic radio resource optimization method are presented, demonstrating significant improvements in throughput. A multi-aspect expanded hypergraph (MAEH) method has been proposed to accurately depict the spatio-temporal features of multi-domain resource allocation in satellite networks (SNs) in [6], along with a two-stage scheme to accomplish resource allocation rapidly with low computational complexity and a high resource/utilization ratio. Prediction is another effective approach in resource allocation and it has been used in rentable LEO satellite communication networks [7]. A long short-term memory (LSTM)-based traffic prediction framework and a task-driven joint resource allocation method have been proposed. In [8], the authors conducted research on overcoming problems in resource scheduling for a large number of accessed users and dynamic wireless environments. They proposed an iterative suboptimal algorithm and an enhanced meta-critic learning (EMCL) algorithm. The simulation results verified the effectiveness of these two algorithms.

The open environment of satellite communication and the scarcity of spectrum resources make spectrum management an important issue in satellite communication systems. A market-driven spectrum allocation technique has been proposed in [9], and the terrestrial agent serves as a seller of spectrum/band. They designed an optimal incentive model to maximize benefits, improve spectrum efficiency, and reduce the agent costs. A cognitive radio (CR)-based spectrum-sharing approach has been used in spectrum management for the satellite-aerial-terrestrial integrated network (SATIN), and a cooperative beamforming strategy has been used to facilitate secure and energy-efficient IoT communications in [10]. In [11], am elastic resource allocation algorithm based on dispersion degree (ERA-DD) has been proposed to handle routing, wavelength, and slot assignment in the satellite-assisted Internet of Things (IoT) communication system. The traffic blocking rate, wavelength utilization, average communication delay, and average initial delay are evaluated through simulations, demonstrating that the traffic blocking rate can be reduced by 32.5% and wavelength utilization can be increased by 1.6%. To cope with the problem of spectral efficiency, user access, and limited uplink transmission resource allocation in satellite IoT, a novel scheduling algorithm that combines simulated annealing and Monte Carlo (SA-MC) has been proposed in [12] to achieve optimal dynamic scheduling. The authors in [13] also proposed an uplink transmission scheme in ultra-dense LEO satellite networks. This scheme employed a two-timescale resource allocation framework to maximize the overall data rate. The small timescale problem was described as a Markov decision process (MDP) to cope with changes in link states relative to satellite movement and rain attenuation, and the large timescale problem was formulated as an integer programming problem to deal with band switching. In [14], the hybrid beamforming, adaptive user scheduling, and resource allocation framework for the terrestrial-satellite network (TSN) has been proposed to mitigate intra-beam and inter-beam interference.

For the multi-beam mega-LEO satellite constellation, mutual interference among satellites and beams limits the implementation of spectra and reduces spectrum efficiency. To solve this problem, an online frequency allocation algorithm based on machine learning (ML) has been proposed in [15]. Under this algorithm, maximizing throughput while constraining interference can be achieved. In [16], the authors also considered interference management in multi-beam satellite communication systems. They use an inter-beam interference coefficient matrix derived from frequency reuse to measure the level of co-channel interference. A novel joint power and bandwidth allocation algorithm has been proposed. The golden-section theory and subgradient iteration have been used to solve the optimization problem. The numerical results demonstrate that the proposed algorithm could maximize capacity and minimize bandwidth utilization variance.

A joint approach to caching, computing, and communication (3C) resource management strategy was presented in [17], and a tapped water-filling algorithm was proposed for the TSN, in which hot air balloons are used as relays between satellites and ground stations. The authors formulated a collaborative resource allocation problem for the TSN and solved it using geometric programming with Taylor series approximation. The 3C resource management model was also studied in [18] to maximize overall system throughput. This problem formulated as a Nash bargaining game and solved by constructing an optimization model. Similarly, resource allocation for the coexistence of satellite and terrestrial wireless communication systems in the same frequency band has been studied in [19,20], and these studies utilize a radio map (RM) to reduce system overhead. In [21], the authors also provided an efficient power and spectrum allocation scheme to minimize inter-component interference while maximizing throughput provided to users. This scheme consists of two simple linear machine learning-based tools and a linear transformation tool.

High-throughput satellites (HTS) are an important composition of future 6G systems. Digital beamforming (DBF) has been proposed, which was combined with power resource allocation to improve the flexibility of HTS systems in [22]. Resource optimization for beamforming (BF) in next-generation satellite communication (SATCOM) with non-orthogonal multiple access (NOMA) has been investigated in [23]. A distributed resource allocation strategy was proposed for BF and payload power resource allocation, and an efficient user scheduling scheme was designed to improve system throughput.

The authors in [24] also studied a soft-defined approach to jointly manage networking, caching, and computing resources in satellite-terrestrial networks (STNs), formulating it into a joint optimization problem which they solved with Q-learning. It has been shown that the performance of STNs can be improved significantly through the joint allocation of networking, caching, and computing resources. The deep reinforcement learning (DRL) method has also been used in resource allocation for STNs. The authors formulated the optimization problem as a Markov decision process to minimize long-term costs in terms of a trade-off between task execution latency and energy consumption [25]. A DRL-based band allocation method in multi-beam satellite communication systems has been proposed in [26] to enhance transmission efficiency with low complexity. In [27], a dynamic user association (DUA) mechanism with task classification was proposed to meet the requirements of load balancing and user task processing in STNs, and a dynamic cell range extension algorithm was developed to adjust the load in terms of resilient backhaul capacity. Furthermore, CR was introduced in frequency reuse and interference management in STNs in [28]. The authors formulated the spectrum optimization problem into a tractable convex optimization form and proposed a closed-form expression for both power allocation and subchannel allocation. In [29], resource allocation has been considered in LEO multi-beam STNs to serve vehicles, and the authors modeled it as a cooperative multi-agent reinforcement learning process where each beam acts as an agent. Centralized training and distributed execution is performed in the training of agents.

Traditional centralized ground control with low invulnerability and large propagation delays cannot adapt to large constellation systems. Hence, distributed resource allocation approaches are preferred, and a lightweight method based on contract theory with on-time interaction has been presented in [30]. To deal with limited power resources and high dynamic characteristics in electromagnetic environments, user distribution and service requirments in LEO satellite communication systems, service, priority, and multi-beam scheduling models have been established in [31]. The authors provided an enhanced artificial bee colony algorithm to improve the ability of quick convergence and global optimization, which has high applicability to the high dynamic characteristics of LEO satellites. The digital video broadcast return channel satellite 2 (DVB-RCS2) standard provides return link specifications for satellite communication. To maximize fairness, a joint superframe and resource allocation problem is decomposed into two-level hierarchical problems, and an iterative algorithm is designed to solve them in [32].

For resource allocation in beam-hopping satellite communication systems, a joint beam-hopping and precoding algorithm has been presented in [33] to achieve resource allocation and intra-cluster interference suppression. The simulation results show that the proposed approach achieves reliable and near-optimal transmission capacity. The authors in [34] studied the combination scheme of beam-hopping and NOMA. They optimized resource allocation by exploiting time-domain flexibility using an optimized BH design and power-domain flexibility using NOMA. In [35], we proposed heuristic and maximum weighted clique (MWC) algorithms to realize the joint scheduling of beams and slots. An auxiliary graph was constructed to find the MWC, where the weights of vertices were assigned while considering not only user/traffic distribution, inter-beam interference, and fairness, but also delay requirements and channel status (path loss and rain attenuation). The simulation results indicate that the approach achieves excellent performance in terms of throughput and degree of user satisfaction. For the interference challenges in multi-LEO constellation coexistence scenarios, the authors in [36] proposed a scheme to tackle this issue through dynamically adjusting the beam bandwidth.

In recent years, some new contributions have been proposed for mega/ultra-dense LEO constellations. In [37], a distributed resource management framework (DRMF) was proposed for a malicious jamming environments for the Internet of Satellites (IoS). This framwork was divided into three sub-problems: a traffic prediction problem, an anti-jamming decision problem, and a resource matching problem. The authors in [38] investigated resource scheduling for satellite–air communication in ultra-dense LEO networks and proposed a framework to deal with the high dynamic characteristics of LEO satellites and aircraft. In [39], a downlink resource scheduling algorithm was presented to achieve a fair allocation of time-frequency resources. The available power, beams, and channel resource allocation in multi-LEO constellation networks has been studied in [40] using a hierarchical multi-agent multi-armed bandit resource allocation for LEO constellations (mmRAL). In [41], a mobility management strategy based on reinforcement learning (RL) has been proposed to minimize handovers and migration delays in mega-LEO satellite constellations. A joint optimization problem of computation offloading and a resource allocation algorithm has been proposed in [42] to minimize service delays and energy consumption. Additionally, the absence of ML techniques for resource management in the MLSCS is noteworthy. The survey in [43] presented the use of ML for radio resource management in mega constellations. In [44], the authors presented a hierarchical and sub-area network control architecture based on the typical characteristics of LEO giant constellations.

All the above works are presented in Table 1. The existing literature on resource management in satellite communication systems or SINs primarily focuses on specific scenarios and the allocation of individual resources. However, there is a lack of research and consideration for a comprehensive network resource management framework for the MLSCS. Therefore, this paper aims to propose a unified resource management architecture to enhance the efficiency of deploying and operating the MLSCS.

## 3. Analysis of Resource Management in the MLSCS

In the MLSCS, the global distribution of SBSs, users, and their services, along with the use of global spectrum resources, greatly affects the scheduling of system resources. How to allocate a greater amount of resources to the places where users and service are concentrated, so as to improve the utilization efficiency of system resources, is the most important issue concerning resource management in the MLSCS.

### 3.1. Influencing Factors Associated with Resource Management in the MLSCS

This study examines the influencing factors associated with resource management in the MLSCS. By identifying and analyzing key technical, operational, and environmental factors, this research aims to provide insights into optimizing resource allocation and management strategies within the MLSCS. Understanding these factors is crucial for enhancing the efficiency and effectiveness of satellite communication systems.

The space segment in the MLSCS consists of hundreds to tens of thousands of LEO satellites operating at altitudes ranging from several hundred to two thousand kilometers. Due to their rapid movement relative to terrestrial users, these satellites create a highly dynamic system characterized by continuous changes in network topology and rapidly shifting coverage areas. Additionally, terminals frequently hand over among different satellites and beams, with service requirements also evolving. This dynamic environment necessitates timely adjustments in resource allocation to effectively adapt to these changes, highlighting the importance of developing responsive resource management strategies.

In the MLSCS, resource scheduling presents a multi-dimensional and multi-objective optimization problem. The multi-dimensional resources include time (or slots), space (beams in multi-beam or beam-hopping satellite systems), frequency (spectrum, channel, or band), and energy (power). The optimization objectives include maximizing throughput (or system capacity), minimizing delay or cost, and maximizing user satisfaction. The MLSCS serves global users; however, the system’s high dynamic characteristics and imbalanced global spectrum resource utilization lead to fluctuating availability of spectra for LEO satellites and low utilization efficiency. Given the limited satellite resources, it is crucial to identify the main factors affecting spectrum scheduling in space systems, which include the following:Regionality of spectrum usage policy. The available spectrum is different in different countries and regions. Therefore, the available spectrum also changes dynamically with the movement of SBSs.Interference from open space. The electromagnetic environment in space is complex and rigorous, and there are various types of intentional and unintentional interference to SBSs and user terminals. In particular, the interference of terrestrial base stations introduced by the application of 5G extended frequency bands has an effect on the resource scheduling of the MLSCS.Protection of GEO satellite spectra by the ITU (International Telecommunication Union). The MLSCS requires multi-satellite collaborative coverage to avoid interference from GEO satellites, especially in mid- and low-latitude areas.Intra-system interference. For Walker satellite constellations, intra-system interference introduced by the aggregation of satellites in the north/south pole areas needs to be optimized and controlled.Transmission capacity of SBSs. Generally, the capacity of a single satellite is limited, so for hotspot areas that need enhanced and urgent services, such as earthquake relief, the regional service volume can increase dramatically in a short time, and it is necessary to allocate more resources to meet users’ demand in a collaborative way. Hence, multi-satellite emergency resource scheduling is required.Onboard processing capability. Due to the limited caching and computing capacity of satellites, it is difficult to realize complex resource scheduling algorithms and massive data exchanges, especially for cross-regional resource scheduling planning, which needs careful consideration.

In summary, the MLSCS faces challenges due to limited satellite capacity, insufficient spectrum, and inadequate service sensing capabilities. These limitations make it difficult to adapt to the complex and rapidly changing electromagnetic environment and evolving service demands. To address these issues, the NCC must uniformly schedule the available spectrum based on the global electromagnetic situation. This scheduling, combined with optimizing resource usage in satellites and planning service areas, aims to enhance the overall performance of the MLSCS.

### 3.2. Primary Challenges in Resource Management

In the design of MLSCS, the Walker constellation is commonly used. The Walker constellation offers several advantages:Stable topology: The movement of all satellites in the constellation is similar, with minimal variation in perturbation and mutual position. This results in a stable constellation shape, which facilitates system control, management, and routing.Near-circular orbit: The Walker constellation adopts a near-circular orbit, where the angular velocity of satellite motion is nearly constant. This characteristic is beneficial for achieving global uniform coverage.

In the MLSCS, resource scheduling is crucial for efficient operation. However, several adverse factors restrict effective resource scheduling. These factors include orbital perturbations, satellite failures, and communication latency, among others. Understanding and mitigating these factors is essential for optimizing the performance of the satellite constellation system.

#### 3.2.1. Non-Uniform Distribution of SBSs

Due to the inherent global configuration of the Walker constellation, the amount of satellite resources that users can see at different latitudes is different. In the multi-satellite coverage simulation shown in Figure 1, the colors represent the number of satellites covering the areas. For example, the green areas are covered by one satellite, while the red areas are covered by six satellites. As shown in this figure, in a typical mega-LEO satellite constellation system, and especially in the polar orbit constellation, users in low-latitude areas can see fewer satellites than in high-latitude areas, especially in the polar areas. With the exception of the orbital seam region, most of the mid- and-low latitude areas are single- or double-covered. With increasing latitude, the phenomenon of multiple coverage becomes more and more serious. The north/south pole areas can evenly be covered by 20 satellites at the same time.

The global distribution of SBSs is highly unbalanced, with mid- and low-latitude areas experiencing limited satellite resources, particularly in service hotspots. In contrast, high-latitude areas experience a concentration of satellite resources, leading to potential intra-system interference due to overlapping coverage and electromagnetic spectrum usage. To mitigate this issue, strategies such as turning off satellite payloads over the polar regions, interference avoidance, and interference elimination techniques should be considered.

To address the unbalanced distribution of SBSs, which leads to insufficient resources in low-latitude areas and intra-system interference in high-latitude areas, it is necessary to adopt elastic and flexible resource scheduling methods. These methods can help optimize satellite resource allocation and mitigate interference, ensuring more efficient and reliable satellite communication services.

#### 3.2.2. Scarce Global Spectrum Resources

The development of satellite communication systems is currently constrained by the scarcity of spectrum resources, which cannot meet the increasing demand for new services and applications. This has led to escalating spectrum congestion and interference issues. Furthermore, competition for spectrum resources between satellite and terrestrial communication systems is intensifying, posing additional challenges for the growth and expansion of both systems.

As communication systems increasingly use higher frequency bands, limited spectrum resources have always been a major constraint on satellite communication systems. As more services and applications adopt these higher frequencies, spectrum congestion is likely to become a significant barrier to the deployment and development of satellite systems. The scarcity of Ka-band and even higher frequency bands has affected the performance of broadband satellite communication systems. Effective spectrum management and planning are crucial for the design of these systems. To address this issue, some researchers have proposed cognitive radio-based satellite spectrum allocation. This approach allows satellite communication to share the spectrum with terrestrial systems under acceptable interference conditions, thereby improving the utilization efficiency of satellite spectrum resources.

On the other hand, limited spectrum resources are divided into several independent frequency bands. To avoid interference, these bands are restricted to specific authorized users. Unauthorized users cannot access these bands even when they are idle. This model ensures that authorized users are not disturbed. However, the exclusivity of these bands in both the time and space domains means that a large number of authorized users have exclusive access. This leads to significant waste and reduced spectrum utilization efficiency.

To address the problem mentioned earlier and reduce spectrum usage limitations, spectrum sensing is used to identify real-time spectrum holes, which can improve spectrum utilization efficiency. Additionally, optimizing spectrum allocation ensures that limited resources are used where they are most needed. This approach helps to make better use of available spectrum and reduces waste.

#### 3.2.3. Non-Uniform Distribution of Service

First of all, the service distribution of MLSCS is non-uniform. For instance, in Figure 2, the users of the Iridium system are concentrated in densely populated, industrial, and commercially developed areas. Conversely, there are very few users in most other areas, such as the sea, deserts, and polar regions. Furthermore, during emergency events, users and traffic will concentrate in a small area within a short period, necessitating capacity enhancement and additional support.

In terrestrial networks, areas with a sharp increase in service traffic can be accommodated by deploying more base stations. However, the capacity of SBSs struggles to meet the high-capacity requirements of emergencies. Furthermore, deploying more SBSs quickly in terrestrial networks is almost impossible. The only viable solution is to optimize resource scheduling within the system to meet these requirements.

#### 3.2.4. Diversified Users and QoS Requirements

A mega-LEO satellite constellation is a satellite system designed to serve global users. It caters to a diverse range of users, each with unique quality of service (QoS) requirements.

Traditional terrestrial users, including fixed stations, portable stations, and relay stations, etc. These stations primarily support land Internet access, land 4G/5G relay services, land emergency communications, polar scientific communications, overseas agency communications for governments and enterprises, and maritime communications. These users are characterized by their limited mobility and lack of cross-regional resource scheduling.

High-mobility, large-span, and planning route users, including Internet access for aircrafts, ocean ships, high-speed railways, and other end users. These users are characterized by high resource demand, high mobility, and a large spatial span. Their movement trajectories can be used to optimize resource scheduling. However, due to their fast movement, combined with the high-speed movement of LEO satellites, a significant dynamic Doppler shift occurs between them and the satellites. This has a substantial impact on communication quality and necessitates frequency correction.

High-mobility, large-span, and on-demand scheduling users. As unmanned aerial vehicle (UAV) technology advances, the transmission of command and control information and reconnaissance data for long-voyage UAVs and missiles relies on space-based communication systems, making them potential users of the MLSCS. These users exhibit strong mobility, random movement, and data transmission capabilities, requiring on-demand (OD) scheduling of resources to meet their dynamic data transmission needs. Additionally, the transmission of command and control information must meet stringent requirements for low bit error rate (BER), low delay, and high security.

Special industry users. Users in special industries are also potential beneficiaries of the MLSCS. Through its global interconnection capabilities, the MLSCS enables these industries to establish their own private networks, such as financial, telecom, medical, and education private networks. Each of these networks serves unique purposes and has varying quality of service (QoS) requirements. To meet these diverse needs, it is necessary to integrate a wireless resource scheduling system. This system optimizes resource allocation across different network systems, ensuring the required delay, rate, reliability, and security. For instance, financial private networks can leverage the MLSCS to enhance transaction security and speed, while telecom private networks can use it to optimize network traffic and reduce latency.

#### 3.2.5. Global Optimization Is Very Difficult

Limited by coverage area and long revisit cycle of SBSs in MLSCS, it is difficult to obtain the global quasi-real-time spectrum situation and service requirements. Hence, the global resource scheduling and optimization cannot be performed.

## 4. Hierarchical Resource Management Architecture

To leverage the low loss and delay of LEO satellites while addressing the challenges posed by their high-dynamic characteristics, the system resource scheduling must be real-time, flexible, and efficient. Optimal resource management schemes must be implemented across multiple domains, including time, space, frequency, and energy (or power). These resource scheduling strategies and approaches differ at various levels within the system, including NCC, SBSs and user terminals. Therefore, it is essential to design a hierarchical resource management architecture that aligns with the distribution and scheduling demands of resources at these different levels. By adopting a hierarchical approach, resource management can be more flexible and efficient compared to considering the entire network as a single entity. Further details on the design and implementation of this hierarchical architecture will be provided in subsequent sections.

Resource management in satellite communication systems is closely tied to the constellation design and coverage scheme. The large bandwidth of these systems allows for a high frequency reuse factor, which helps to minimize co-channel interference (CCI) in overlapping coverage areas of SBSs. In multi-beam or beam-hopping systems, the spectrum is efficiently multiplexed among beams to improve spectrum efficiency. Spot-beams, which are primarily responsible for data transmission in satellite communication systems, require a significant amount of bandwidth. However, because spot-beams are spatially isolated, a smaller frequency reuse factor can be used among them to enhance spectral efficiency. Furthermore, multi-beam or beam-hopping technology can be employed to increase the spectrum reuse factor and reduce inter-beam CCI, further optimizing resource management in satellite communication systems.

Compared to traditional GEO and multi-beam satellite communication systems, resource scheduling in the MLSCS is significantly more complex. The limitations of satellite resources, such as power, bandwidth, and beam, require a careful balance between providing global on-demand service and optimizing resource utilization. Additionally, the dynamic nature of coverage areas in MLSCS necessitates an adaptive resource scheduling approach. Based on the service data flow and resource distribution within the system, wireless resource scheduling in the MLSCS can be divided into three levels. These levels may include high-level strategic scheduling, mid-level tactical scheduling, and low-level operational scheduling, each with its own unique challenges and considerations.

Level 1: To ensure efficient and effective utilization of satellite resources, it is necessary to consider the division of satellite service areas and the available spectrum resource planning. By leveraging these elements, we can guide satellites to carry out multi-satellite service coordination, ensuring that they operate in a complementary and synergistic manner. Furthermore, by establishing a unified global collaborative service system, we can enable seamless communication and data sharing across different satellite networks, promoting a more integrated and efficient satellite communication ecosystem.Level 2: To ensure optimal utilization of satellite resources, we allocate them to user terminals based on their specific needs and requirements. This allocation process involves guiding user terminals to access SBSs that are strategically located to provide the best possible coverage and service. Once user terminals are connected to SBSs, data interaction between them is realized through a secure and efficient communication protocol. This protocol ensures that data are transmitted and received in a timely and accurate manner, promoting seamless communication and data sharing across the satellite network.Level 3: This level schedules user data for various service types, such as real-time, best-effort, and guaranteed delivery services. Subsequently, we will optimize the transmission of these data through advanced scheduling algorithms and network protocols to achieve robust QoS guarantees.

Similar to terrestrial mobile base stations (BSs), SBSs are the primary entities responsible for resource management in the MLSCS. They directly schedule and employ resources such as time slots, directional beams, radio spectrum, and power to provide access to services. Furthermore, unlike terrestrial mobile communication systems, the MLSCS exhibits the following characteristics:Compared with terrestrial mobile base stations, LEO satellites have distinct characteristics. They move quickly and cover large areas. However, LEO satellites have a very short residence time over any given location, meaning their service area changes rapidly. This leads to frequent changes in the users they can serve.Service demands are not uniformly distributed across all areas. In hotspots or emergent situations, it is necessary to implement multi-satellite collaboration to provide enhanced services, such as increased capacity, improved coverage, or larger data rates.User handover between beams (the areas of coverage for individual satellite signals) and satellites is not solely determined by signal strength measurements. It also involves multi-satellite resource negotiation, which takes into account various factors such as satellite availability, capacity, and service priorities.Dynamic change in the areas covered by LEO satellites makes status data, such as spatial spectrum and service demands, incomplete or non-real-time (NRT), which makes resource optimization and scheduling more difficult.Due to their short residence time, SBSs have insufficient cognitive ability regarding propagation circumstances and service situations. Therefore, it is necessary to coordinate multi-satellite cognition to support system resource scheduling.The most critical problem is that SBSs have limited resources, including computing, caching, and transmission resources. To achieve optimal system resource scheduling, terrestrial assistance is necessary. However, if NCC pre-schedules all resources, the resulting processing delays and complexity would be too high to meet real-time service demands.

Based on the above analysis, the hierarchical resource management architecture of MLSCS is proposed here, to achieve optimal scheduling of system resources and fulfill real-time service demands. The radio resource scheduling architecture within MLSCS is divided into three layers, as shown in Figure 3.

Radio resource scheduling center (RRSC). It is responsible for system resource scheduling, integrating several functions: (1) It collects network situation information, including details on global telecommunication policies and available resources. (2) Using historical data, it optimizes service area planning and global spectrum scheduling. (3) It schedules SBSs for hotspot and emergent areas to ensure coverage and capacity. (4) It configures the resource scheduling strategy and guides SBSs in scheduling resources efficiently.Satellite resource scheduling (SRS). SRS is located at the SBSs and is user-oriented. Taking the configuration of the RRSC as constraints, the beams, slots, spectrum and power (modulation parameters) are scheduled at SBSs, considering user demand and capacity to provide access services for users. At the same time, the downlink data are scheduled to realize the dissemination of user data according to priority.User service scheduling (USS). The scheduling of terminal service data occurs at user terminals. Based on their service demands, user terminals submit resource requests to SBSs. The SBSs, in turn, prioritize and manage links and data based on the QoS requirements, which include factors such as latency, bandwidth, and reliability. This management ensures that different service types, such as voice, video, and data, receive the necessary resources to meet their specific demands, thereby achieving resource optimization for user terminals.

According to the three-level management architecture, the RRSC is responsible for planning satellite service areas and spectrum resources based on the global distribution of available spectrum, service demands, satellite loads, and other relevant data. Based on these planning data, SBSs implement the scheduling of beams, frequency, time slots, and power for users. Meanwhile, the user service scheduling module primarily deals with the priority adaptation of different service data to ensure orderly access. The hierarchical resource management architecture achieves functional hierarchy and optimal scheduling of radio resources by decoupling resource scheduling and processing through global control at the RRSC, area optimization at the SBSs, and service stream control at user terminals. This functional partitioning allows the RRSC to shield the influence of dynamic changes (such as varying service demands and satellite loads) on SBSs, thereby simplifying the resource scheduling demands of SBSs.

## 5. Detailed Elaboration of Each Layer

This section introduces the roles and functions of the RRSC, SRSC, and the USS. It also presents some procedures for realizing these functions. By understanding the functions of each component and the procedures used to achieve them, readers can gain a better understanding of how the overall system operates.

### 5.1. Functions of Radio Resource Scheduling Center

MLSCS provides high-reliability, low-delay services to user terminals. However, the strong mobility of LEO satellites and the imbalance of resource usage worldwide cause the available spectrum and coverage area of SBSs to change dynamically. Therefore, the RRSC is primarily responsible for resource planning for the entire constellation, including global spectrum resource scheduling and service area planning for all SBSs.

The primary function of the RRSC in MLSCS is to analyze spectrum usage and service evolution patterns by using machine learning. This analysis considers historical data (such as user distribution, service distribution, and interference), current usage status (including spectrum usage, interference status, real-time service distribution, and signal bursts), and national/regional spectrum usage policies. Grids can be used as the basic unit for planning SBSs’ service areas, and game theory-based spectrum planning is employed to guide SBSs’ frequency usage behavior. In addition to normal management, mandatory scheduling strategies are needed for service enhancement in hotspot and emergency rescue areas. To address the problem of temporal-spatial mismatch between global service and resource, a multi-satellite collaborative support strategy will be used at multiple levels, such as SBSs and beams. The specific functions of the RRSC in MLSCS are as follows (Figure 4):

Interference avoidance: ITU regulations grant GEO satellites priority in the use of spectrum compared to LEO satellites, meaning that the spectrum used by GEO satellites is protected. In middle- and-low latitude areas, however, common line-of-sight (CoLoS) interference may occur between GEO and LEO satellites. To avoid this interference and ensure compliance with ITU regulations, the MLSCS must address two key issues. First, user beam pointing must be prohibited towards GEO satellites to prevent interference. Second, feeder link interference avoidance measures must be implemented to protect GEO satellites from interference from LEO satellites. Addressing these issues will be critical to the successful implementation of the MLSCS.

Scheduling of feeder links: To ensure that the feeder link connection is optimized and avoids interference from GEO satellites, we first perform calculations for GEO interference avoidance. Based on these calculations, we generate a plan to guide the SBSs and gateways in completing the feeder link scheduling. This plan includes specific instructions for beam pointing of the feeder link. The beam pointings are then generated and distributed to the corresponding control units, which are responsible for real-time feeder link scheduling. Following this process, we can ensure that the feeder link connection is optimized and compliant with GEO interference avoidance requirements.

Interference management: Combined with system sensing data, the RRSC system supports global interference cognition and management. Terrestrial voluntary and involuntary interference in various regions can potentially impact the performance of the MLSCS, especially with the expansion of the 5G frequency band. To address this, the RRSC collects spectrum sensing data from SBSs and user terminals worldwide equipped with spectrum sensing modules. These data help construct a comprehensive global spectrum situation. By mining the temporal-spatial laws of interference from these data, the RRSC optimizes resource planning. Additionally, multi-satellite and multi-terminal sensing data are integrated to achieve interference cognition and enhance quasi-real-time spectrum scheduling, ultimately improving the system’s service quality.

Global spectrum management: The mega-LEO satellite constellation system aims to provide comprehensive communication services to global users. However, spectrum availability varies across different regions, and spectrum usage policies and requirements also differ. Additionally, the dynamic movement of SBSs further complicates spectrum management, as the available spectrum changes with their location. To address these challenges, the RRSC system implements global spectrum management and planning for SBSs’ payloads. This is achieved by integrating spectrum sensing data with local radio resource usage policies, ensuring efficient and compliant spectrum utilization across the globe.

Service area planning: Based on interference avoidance and management strategies, as well as the available resources of SBSs, the system dynamically plans service areas to meet global service demands. To configure each SBS’s payload, the Simple Network Management Protocol (SNMP) is utilized. By combining mobility management, the system achieves efficient scheduling of global resources. This approach optimizes service area planning in multi-satellite overlapping areas and schedules user access to meet user service demands. Additionally, the system supports multi-satellite resource collaborative scheduling (MSRCS), which is designed to meet the enhanced service requirements of hotspots and emergency services, such as earthquake relief. Through MSRCS, the system maximizes on-demand service capabilities and realizes space service capability scheduling, ensuring that resources are allocated effectively and efficiently.

Payload management of SBSs: The convergence of small base stations (SBSs) over the north/south pole areas results in overlapping interference, including signaling beams, which can affect the scheduling and planning of various resources. To eliminate this intra-system interference, the RRSC should configure regular on-off control for space payloads in these areas.

### 5.2. Functions of Space Base Stations

At RRSC, service area planning and spectrum scheduling dictate how SBSs allocate resources such as slots, beams (and their direction), frequencies, and power. These allocations are based on a fair principle, ensuring that the SBSs can meet the diverse service data needs of various user terminals. The specific functions of resource scheduling at SBSs are as follows:

Service area and radio resource configuration: The RRSC coordinates with SBSs to optimize network performance, directs SBSs to adjust their service areas—expanding or contracting coverage based on demand and congestion—and configuresradio resources according to its comprehensive service area and spectrum planning, which involves allocating specific frequency bands to minimize interference and maximize network efficiency.

User clustering: User clustering is carried out to optimize resource allocation in multi-beam frequency multiplexing or beam-hopping scheduling scenarios. The process considers the service capacity of each beam and the capacity requests of users within the service area of an SBS. Users in a cluster share the same beam resource, while inter-cluster resources are coordinated based on service demands. The number of users in a cluster can be adjusted according to demand changes across different scheduling cycles.

Beam management: Based on the principles of fairness and capacity maximization, SBSs have the functions of beamforming and beam-hopping pattern calculation. Beamforming helps to direct signals towards specific users, while beam-hopping patterns are calculated to suppress mutual interference among beams. These functions also enable SBSs to schedule user access slots efficiently.

User parameter management: Link parameter scheduling is carried out for user terminals, considering the terminal type and link status. The transmission parameters for each user link include power, frequency, and modulation and encoding (MODCOD) sets. These parameters are optimized to accommodate changes in the link status and improve user link capacity. By optimizing these parameters, we can ensure better performance and reduce errors in transmission.

### 5.3. Functions of User Terminals

User terminals have several functions, including user data management, access and band request, and receiving satellite broadcasting or control information. In response to this information, user terminals perform various actions. The access network relies heavily on access protocols, which are divided into two main types: random access and on-demand access. Random access allows user terminals to initiate communication without prior coordination, while on-demand access requires coordination between the user terminal and the network before communication can begin.

#### 5.3.1. User Access Stratigies and Procedures

In the MLSCS, the signaling channel employs a scheme that combines on-demand access (DA) and random access (RA) generally, while the service channel uses only on-demand access. Consequently, the access methods and procedures for the signaling channel and service channel differ.

Signaling channel access procedure: Signaling channel access combines DA and RA. Different user types and service types require different access methods, with corresponding access procedures. Figure 5 illustrates the access procedure for the signaling channel. To apply for the signaling channel, the user first submits a request to SBS. Based on the user and service type, the SBS determines whether to use random or on-demand access. For random access, the SBS checks for any idle random access channels. If an idle channel is available, it is allocated to the user. If no idle channels are available, the access procedure is restarted after a timeout. For on-demand access, the SBS checks the occupancy of reserved or authored channels. If an idle channel is available, it is allocated to the user. If no idle channels are available, the user waits until a channel becomes idle and is then allocated for access.

Service channel access procedure: The service channel access procedure is illustrated in Figure 6. Spectrum/channel allocation is based on service priority to ensure that high-priority and online users’ services can be successfully accomplished and that the QoS is guaranteed. When channel resources are limited, low-priority users or users with low-priority services must compete for access to the system. The user access order is determined based on their priority and the results of any games or competitions that may be used to allocate the limited resources.

#### 5.3.2. Random Access Stratety

Opportunistic spectrum access is a promising technology for users, especially low-priority users accessing SBSs. When users’ service streams arrive, they enter a random access status. When there is an opportunity, the SBS allocates time-and-frequency resources to the users for their service accesses.

Another approach is random access based on NOMA protocols. In this method, users transmit signals to the SBS on the same time-and-frequency resource block. The SBS distinguishes these signals using multi-user detection (MUD), which relies on unique signatures for each user. The MUD technologies include successive interference cancellation (SIC), despreading, descrambling, deinterleaving, compressed sensing, and machine learning, etc.

In previous satellite communication systems, grant-based random access protocols with multiple handshakes were commonly used. However, as the number of users increases, grant-free random access protocols are gradually being introduced to reduce handshake overhead and access delay. Figure 6 illustrates the grant-based random access procedure, while Figure 7 shows the grant-free random access procedure.

## 6. Approaches and Algorithms

### 6.1. Global Spectrum Planning

Spectrum is the basic but scarce resource of wireless communication systems. Global spectrum planning is the basis and premise of resource utilization in MLSCS, and also an important approach of efficient utilization of spectrum resources. In the long term, the integration of spectrum from space to ground and providing differentiated services are the important direction and trend. Hence, global spectrum planning is an important function of RRSC. The spectrum usage of SBSs and beams is analyzed, predicted, and planned at the RRSC based on spectrum usage requirements, the global spectrum situation, historical spectrum usage data, user/service distribution and priority, service area planning results, etc.

The spectrum planning system encompasses various features such as spectrum situation sensing, frequency allocation, conflict detection and elimination, interference detection and avoidance, efficiency evaluation, visualization of spectrum planning insights, a spectrum database, and other modes. The functional model of the spectrum planning system is illustrated in Figure 8 and is divided into six primary modules:

#### 6.1.1. Planning Input

The spectrum planning module requires various inputs to function effectively. These inputs include the following aspects:

System frequency band: The system design determines the authorized frequency band and some public frequency bands. These frequency bands, in turn, establish the range for frequency planning.

Regional forbidden frequency band: Each region specifies frequency bands that cannot be used by other wireless systems, based on their unique needs and spectrum usage. When SBSs communicate with users in these areas, it is essential to avoid using these forbidden frequency bands.

Historical spectrum data: The recorded and stored historical spectrum data can be used to identify patterns of temporal and spatial usage through data mining techniques. These patterns help predict future spectrum usage, which in turn informs spectrum allocation and management plans.

Historical interference data: The recorded and stored historical interference data can be used to identify patterns of temporal and spatial interference. By analyzing these patterns, future interference events can be predicted, and the predicted results can inform spectrum planning.

Historical user/service data: The temporal-spatial distribution of users and services is recorded and stored. Through data mining, the temporal-spatial laws governing the distribution of users and services are obtained. These laws are then used to predict the temporal-spatial distribution of active users and services in the future, serving as input for spectrum planning.

Spectrum usage status: Spectrum planning uses two key types of input. First, it incorporates the current spectrum usage status of various areas and the space electromagnetic environment data sensed and collected by spectrum monitoring equipment. Second, the historical spectrum usage status provides essential data for mining and predicting future patterns or trends in spectrum usage.

#### 6.1.2. Planning Constraints

In addition to the factors mentioned above, there are several key considerations that should be taken into account during spectrum planning.

Demand constraints: User demand constraints primarily pertain to users’ QoS requirements, such as bandwidth, delay, and BER.

Interference constraints: Interference constraints include several factors that must be considered during system spectrum planning. These include interference avoidance with GEO satellites, CCI outside the system, CCI among SBSs within the system, and CCI among beams of an SBS. Specific management strategies can be employed to address these constraints; for example, SBS reselection can be used for interference avoidance with GEO satellites, and beam scheduling can manage inter-beam interference within an SBS. Additionally, inter-satellite interference within the system and CCI outside the system are key factors that must be carefully considered during spectrum planning.

Link status constraints: In satellite communication systems, the influence of rain attenuation must be carefully considered when establishing connections between SBSs (small base stations) and users/gateways. SBSs with shorter rain paths and lower attenuation levels are typically better choices for users, as they can provide more reliable and consistent connectivity. Furthermore, the selection of SBSs has a direct impact on spectrum planning, as it influences factors such as frequency allocation, signal strength, and overall network performance.

#### 6.1.3. Data Mining

This module employs various machine learning algorithms, including LSTM networks, correlation calculation methods, causal regression models, and anomaly detection techniques. These algorithms analyze historical data to uncover the temporal and spatial patterns or ‘laws’ governing spectrum usage.

#### 6.1.4. Temporal-Spatial Law of Spectrum Usage

This module stores the patterns of spectrum usage discovered by the data mining module. It records and stores these patterns in terms of time, space, frequency, and power dimensions for a specified time period, area, and frequency band. The level of detail or ‘granularity’ in each dimension is determined by the system planning parameters.

#### 6.1.5. Spectrum Planning

The spectrum planning module uses historical data, status data, constraints, and identified patterns or trends in spectrum usage to plan spectrum usage for a future period across the entire system. This planning is accomplished using various methods, including graph theory, genetic algorithms, and machine learning.

#### 6.1.6. Planning Output

The output module is responsible for storing the spectrum planning results in a database or file system, typically in structured formats such as tables or graphs. Additionally, the module displays the results through dashboards, reports, or other visualizations, providing real-time or periodic insights into the planned spectrum usage. The software architecture of the spectrum planning system is divided into four layers according to the idea of “platform + application”, named as infrastructure layer, general service layer, service logic layer, and application service layer from bottom to top, as shown in Figure 9.

Infrastructure layer. This layer provides computing services, storage services, data transmission, and basic database resources for service and application, while providing physical resource management and scheduling functions.General service layer. It realizes the basic operations of spectrum data, such as collection, integration, conversion, and sharing. It also provides unified spectrum data to provide application-oriented general service support.Service logic layer. By integrating the service bus with the basic support provided by the general service layer, the system realizes spectrum planning and auxiliary decision-making functions, including radio wave propagation prediction and electromagnetic compatibility analysis.Application service layer. Based on the service demands of user terminals at various levels, the network system offers tailored data and computing services. It conducts spectrum planning and sets environmental parameters to ensure optimal performance. Additionally, the system provides a user-friendly interface (UI) for ease of operation and a simulation display interface for visualizing the processes.

### 6.2. Game-Based Spectrum Planning

Multi-satellite resource management is a core task for the RRSC. The RRSC schedules and allocates spectrum resources to SBSs based on the current spectrum usage and network topology. The SBSs then configure these allocated resources among the beams and user terminals. However, traditional fixed spectrum scheduling strategies cannot adapt to dynamic changes in system topology, spectrum usage, and user demand. To address this, dynamic optimization of spectrum resources throughout the network is required. One effective approach is to allocate spectrum resources using pricing mechanisms and competitive bidding, similarly to the commodity market. For this reason, game theory has been introduced into the resource scheduling of wireless communication systems to provide a framework for efficient resource allocation.

In our architecture, the competition and game for spectrum resources among SBSs are managed by the RRSC. SBSs apply to RRSC for usage rights of spectrum based on their individual demands. The NCC formulates the rules and regulations for spectrum usage and sets prices for the spectrum according to the resource allocation strategy and pricing principles. Each SBS then participates in a game-theoretic bidding process to secure usage rights of the spectrum based on these rules and prices. The optimization goal of this game and bidding process is to maximize the utility function of RRSC, which represents the overall efficiency and performance of the network.

After the RRSC prices, the spectrum resource as unit price *p* and $q_{i}$ is the resource requirement of the *i*-th SBS. Suppose that the resource requirment set of *M* SBSs is $Q=\{q_{1},q_{2},\cdots ,q_{M}\}$ and the total demand is $Q=\cup_{i=1}^{M}\left\{q_{i}\right\}$.

Generally, the acquisition of the equilibrium solution of the single-master multi-slaver Stackelberg game between the RRSC and SBSs is completed in two stages. In phase 1, the RRSC firstly announces its pricing strategy *p* to all SBSs. The SBSs determine their resource demand strategy *q* according to the received pricing strategy *p*. In phase 2, the RRSC adjusts its price to achieve larger utility after it obtains the resource demands Q of the SBSs.

The utility function at the RRSC is equal to the difference between its benefit and cost, i.e.,(1)G(p,Q)=(p−c)Q
where *c* is the unit cost of the spectrum and *Q* is volume of resource demand when using Q strategy. The revenue of the RRSC is influenced not only by its own price strategy, but also by the demand of the SBSs. When the RRSC prices too high, the demands of the SBSs reduce and leave more spectrum idle. Contrarily, a low price at the RRSC results in excessive demand and a large network load, which reduce the user experience and system revenue. Therefore, the RRSC is apt to set a reasonable price to maximize its own revenue during the game. Assuming that the optimal price strategy of the RRSC is $p^{*}$, and its maximum revenue is satisfied by $max\{G\left(p^{*},Q^{*}\right)\}$, where $Q^{*}$ is the optimal resource demand strategy of all the SBSs, then $\left(p^{*},Q^{*}\right)$ represents an equilibrium point of the single-master multi-slaver Stackelberg game. The task of the RRSC is to provide a reasonable pricing mechanism so as to lay a price foundation for the game of spectrum for the whole network.

In the non-cooperative game problem, when all SBSs have reached a state of no further strategy changes, and if any SBS unilaterally changes its strategy, it will not result in an increase in overall system revenue, and the utility of each SBS will be maximized. This state is known as the Nash equilibrium, which can be expressed as follows:(2)$$G_{i}\left(p^{*},q_{i}^{*},q_{-i}^{*}\right)\geq G_{i}\left(p^{*},q_{i},q_{-i}^{*}\right)$$
where $G_{i}$ and $q_{i}^{*}$ are the revenue and optimal demand strategy of the *i*-th SBS, respectively, and $q_{-i}^{*}$ is the optimal demand strategy of SBSs other than the *i*-th SBS.

The first-order partial derivative of any *i*-th SBS’s revenue function can be achieved as follows:(3)$$\frac{\partial G_{i}\left(p,q_{i},q_{-i}\right)}{\partial q_{i}}=\frac{a}{1+q_{i}}-\frac{b}{Q}-\frac{Pr^{*}\left(SBS_{i}\right)}{{\left(C-q_{i}\right)}^{2}}$$
where *a* is a numerical value between [1,2], which is related to the user’s QoS requirement (the larger the *a* value, the higher the user’s QoS requirement, the more urgent its demand for physical resources, and the more it pays to the RRSC), and *b* is a constant greater than zero. *C* represents the maximum amount that the RRSC can provide of a certain resource. $Pr^{*}\left(SBS_{i}\right)$ is the normalization importance quantification of the *i*-th SBS that is carrying out a service, which can be achieved as in [45].

The second-order partial derivative of any *i*-th SBS’s revenue function can be achieved as follows:(4)$$\frac{\partial^{2}G_{i}\left(p,q_{i},q_{-i}\right)}{\partial q_{i}^{2}}=\frac{a}{1+q_{i}}-\frac{Pr^{*}\left(SBS_{i}\right)}{{\left(C-q_{i}\right)}^{3}}<0$$

The second order partial derivative is less than 0 so that the revenue function of an arbitrary SBS is strictly concave, which ensures the existence of the Nash equilibrium. Therefore, the revenue function $G_{i}\left(p,q_{i},q_{-i}\right)$ of the SBS exists at the non-cooperative game equilibrium point $q^{*}$ for the price strategy *p* given by the RRSC. Reverse induction is a commonly used method to solve for the optimal strategy in Stackelberg games. However, this method typically requires participants to have complete information about the strategies and payoffs of all other participants. In the context of the above problem, where such complete information may not be available, reverse induction is not suitable. Therefore, we use a distributed iterative method to find the Nash equilibrium in Stackelberg games.

Assuming that the RRSC announces a pricing strategy at time *t*, SBSs need to adjust their resource demand strategies to achieve a Nash equilibrium solution. In this context, a Nash equilibrium refers to a set of strategies where no SBS can improve its utility by unilaterally changing its strategy. The resource demand strategy of each SBS directly determines its revenue function, which is modeled using a simple dynamic approach. Specifically, the change rate of an SBS’s resource demand is proportional to the first-order deviation of its revenue function from its optimal level. This relationship is used to model the dynamic adjustment of resource demand strategies by SBSs in response to the pricing strategy announced by the RRSC.(5)$$\Delta q_{i}=\frac{\partial G_{i}\left(p,q_{i},q_{-i}\right)}{\partial q_{i}}$$

Furthermore, to model the dynamic adjustment of resource demand strategies by SBSs over time, we introduce a resource demand strategy iteration equation. This equation describes the change in the resource demand strategy of the *i*-th SBS between the current moment *t* and the next moment t+1, which can be expressed as follows:(6)$$q_{i}\left(t+1\right)=q_{i}\left(t\right)+\theta_{i}\Delta q_{i}$$
where $\theta_{i}$ represents the resource demand’s adjustment step (or learning rate) for the *i*-th SBS. This step size will affect the speed of convergence towards the equilibrium state. An important property of the system is that the revenue function is concave, which ensures that the iterative algorithm used to update resource demands can converge to the Nash equilibrium of the game. The Nash equilibrium is a stable state where no SBS has an incentive to unilaterally change its resource demand given the resource demands of the other SBSs.

After SBSs reach the Nash equilibrium, the RRSC adjusts its own price through iteration according to the demand strategy distribution of the SBSs. The changing rate of the RRSC’s price is expressed by the After the SBSs reach the Nash equilibrium, the RRSC adjusts its price through an iterative process based on the demand strategy distribution of the SBSs. The rate of change in the RRSC’s price is determined by the marginal utility, a concept from microeconomics that represents the additional benefit gained from consuming one more unit of a good or service. The specific equation used to iterate the RRSC’s price is as follows:(7)$$p\left(t+1\right)=p\left(t\right)+\mu \frac{\partial G(p(t),q(t))}{\partial p(t)}$$
where μ, which is greater than 0, represents the adjustment step of the RRSC’s pricing strategy. This parameter plays a crucial role in determining the convergence rate, which is the speed at which the system reaches the equilibrium state. In other words, the value of μ affects how quickly the pricing strategy adjusts to achieve the desired equilibrium.

The marginal utility of the RRSC can be determined by calculating the change in revenue resulting from a change in price. This can be mathematically expressed as follows:(8)$$\frac{\partial G(p(t),q(t))}{\partial p(t)}\approx \frac{G(\ldots ,p(t)+\epsilon ,\ldots )-G(\ldots ,p(t)-\epsilon ,\ldots )}{2\epsilon }$$
where ϵ is a very small value to calculate partial derivatives.

Through the iterative process, the system enables SBSs and the RRSC to achieve an optimal resource demand strategy $Q^{*}$ and price strategy $p^{*}$, respectively. Since both SBSs and the RRSC reach the Nash equilibrium, the two-stage single-master multi-slave Stackelberg game achieves the Nash equilibrium $\left(p^{*},Q^{*}\right)$.

The iteration process is as follows:

(1) At every moment *t*, the RRSC adjusts its pricing strategy based on the marginal utility derived from Equations (7) and (8).

(2) Once the SBSs become aware of the RRSC’s pricing strategy, they adjust their demand strategy based on Equations (5) and (6) until their utility reaches a maximum value and the total revenue of all the SBSs reaches an equilibrium.

(3) If the revenue of the RRSC reaches an equilibrium, the iteration process stops. Otherwise, at the next time step t+1, the RRSC adjusts its pricing strategy based on the resource demand strategy of the SBSs and repeats step 1.

### 6.3. Service Area Planning

Service area planning is a crucial function of the RRSC. This process is influenced by numerous factors, including grid partition, the distribution of communication services, interference avoidance with GEO satellites, and the management of intra-system and inter-system interference. Additionally, it considers the residence time and propagation distance between SBSs and grids (or users), the available resources of SBSs, and the continuity and heritability of service areas.

Given the dynamic nature of LEO satellites, particularly their rapid orbital changes and positional shifts, and considering the unique technical features and application demands of the MLSCS, several key factors will influence service area planning and resource scheduling. These factors are outlined below:The high-speed movement of SBSs causes their coverage area to change constantly, which, in turn, affects the planning of their service areas.The non-uniform distribution of users, changing user traffic, and high-speed movement of dynamic users (such as those on high-speed trains and aircraft) lead to rapid and uneven changes in service distribution. These factors pose significant challenges to the dynamic scheduling of service areas and resources.Due to different spectrum usage policies and statuses in various regions, the available spectrum resources for SBSs are also dynamically changing.The high-speed movement of SBSs constantly changes the constellation topology and the connection relationships between users/gateways and SBSs. This results in frequent handovers of users/gateways among different SBSs and beams.

To successfully complete the service area planning and resource scheduling in the MLSCS, several key challenges must be addressed. By tackling these issues, we can ensure that our planning and scheduling efforts are effective and efficient.

How can we efficiently calculate and dynamically schedule service areas in response to changes in SBSs’ positions and available resources, constellation topology, and user and service distribution? To address this challenge, we need to develop methods or algorithms that can quickly adapt to these dynamic factors and optimize service area planning.How can we reduce user handovers among different SBSs and beams while ensuring extended connectivity time for users and reasonable scheduling of service areas and resources? To achieve this, we need to develop methods or algorithms that can optimize the placement and scheduling of SBSs and beams to minimize handovers and maximize user satisfaction.How can we ensure real-time and reliable space–ground link connections while minimizing the costs or delays associated with handovers? To address this challenge, we could consider using advanced communication protocols, optimizing network topology, or implementing efficient handover algorithms. These solutions could help ensure that users maintain continuous and reliable connections while minimizing the disruptions, delays, and overheads caused by handovers.

Given the challenges discussed above, the RRSC should follow the following principles when planning its service areas:The SBS must cover the grids in its service area. These grids are small areas divided on the ground based on longitude and latitude, serving as the basic unit for satellite service area planning. The SBS’s service area must be entirely within the coverage area of a larger satellite service provider; otherwise, it cannot meet the communication conditions and provide service to users within the grids.Interference avoidance for GEO satellites is crucial. The connections between users and SBSs must be kept away from the geostationary orbit to prevent interference with GEO satellites. If a connection between a grid and an SBS passes too close to the geostationary orbit, grid users must be handed over to adjacent SBSs to maintain communication without interference.Ensuring sufficient resources in SBSs is crucial for providing effective and reliable service to users in their service areas. To maintain an essential balance of resource usage among SBSs, it is important to allocate resources in a way that ensures each SBS has a certain margin. When available resources are insufficient, user priority scheduling can be implemented to meet user requirements in batches. This scheduling method allows SBSs to prioritize users based on certain criteria, such as their importance or urgency, and allocate resources accordingly.To ensure the relative stability of user connections and reduce handover overheads, it is important for users to select an SBS with the longest possible residual time. This helps reduce the frequency of user handovers among different SBSs and beams, which can be costly and disruptive. By selecting an SBS with a longer residual time, users can maintain a stable connection and avoid unnecessary handovers.To improve the reliability and efficiency of data transmission, it is important to minimize the propagation distance between an SBS and the grid. This helps reduce propagation loss, which is the reduction in signal strength that occurs as the signal travels through space. By selecting an SBS closer to the grid, users can experience better data transmission performance and reduced latency. Additionally, minimizing propagation distance can help improve the overall capacity of the network by allowing more users to connect to the SBS without experiencing significant signal degradation.The RRSC should not only focus on multi-satellite cooperative services in hotspot areas, but also plan for service satellites in sparsely populated and low-traffic areas to ensure that SBS resources can be allocated to these areas.

Therefore, the service area scheduling algorithm in the MLSCS must consider various factors, such as coverage, interference avoidance with GEO satellites, residual time, propagation distance, and so on. That is to say, the service area scheduling algorithm should contain the following sub-algorithms: the SBS coverage area calculation algorithm, the interference avoidance calculation algorithm, the residual time calculation algorithm, the propagation distance/loss algorithm, and the rain attenuation calculation algorithm, etc.

### 6.4. Beam Scheduling for Beam-Hopping

Beam-hopping technology is increasingly used in LEO broadband satellite communication systems, such as Starlink, to improve performance. Beam-hopping involves using a limited number of beams to achieve the coverage of a traditional multi-beam satellite system. This is based on time slicing technology, where beams are activated and deactivated in a sequence to cover specific areas of the satellite’s service area. In a beam-hopping slot, all or part of the satellite beams are in the working state, lighting up cells in the satellite service area based on demand. This concept—“where there is a demand, the beam goes there”—allows beam-hopping to better adapt to unbalanced service demands compared to traditional multi-beam technology. By dynamically allocating beams based on demand, beam-hopping can optimize network resources and improve overall system performance.

Beam-hopping technology uses spatial isolation among the working beams to reduce CCI. However, it does not completely solve the interference problem. Specifically, if the angle between two beams is small, the sidelobes of the beams can cause interference, affecting the signal quality. To control this interference, a beam scheduling algorithm is designed. This algorithm optimizes the activation and deactivation of beams to avoid the impact of inter-beam interference on communication quality. By carefully scheduling the beams, the algorithm can improve system throughput and ensure reliable communication. In summary, beam-hopping technology, combined with a well-designed beam scheduling algorithm, can effectively reduce CCI and improve system performance in LEO broadband satellite communication systems.

The beam scheduling algorithm must take into account several constraints to ensure efficient and effective resource allocation, which involves the following areas:The ephemeris of SBS. The location of an SBS, which includes its longitude, latitude, and altitude, or XYZ coordinates in the Earth-centered Earth-fixed (ECEF) coordinate system, is crucial for determining the area it can cover. The SBS’s position is directly related to the grids that it can serve, and these grids are associated with specific users and services. To generate an effective beam-hopping pattern, it is necessary to calculate the number of slots required for each cell within the SBS’s service area. This calculation is based on the distribution of services within the area. Once the number of slots required for each cell is determined, the slots can be allocated to each cell according to its needs. By understanding the relationship between the SBS’s location and the grids it can cover, as well as the process of calculating and allocating slots for each cell, we can ensure that the beam-hopping pattern is optimized to meet the needs of users and services within the SBS’s service area.Service area planning. The service area range of the space base station, meaning the location area it contains, is determined by the scheduling of the service area of the NCC. When calculating the beam-hopping pattern, the space base station only needs to pay attention to the users and services within its service area.User and service distribution. To generate the beam-hopping pattern, the SBS first assesses the number of slots required for each cell within its service area. Slots refer to the time intervals during which the SBS can communicate with users. The SBS then allocates the required slots to each cell, taking into account the distribution of users and services within that cell. After allocating the slots, the SBS schedules the beams according to the beam-hopping pattern. This scheduling process involves determining the sequence and timing of the beams, as well as the power and frequency of each beam. By carefully scheduling the beams, the SBS can ensure efficient and effective communication with users in its service area.The distribution of the cells. In the MLSCS, the size and shape of each cell are also associated with the grids that are used to organize the system. Grids are a set of imaginary lines or boundaries that divide the area covered by the satellite into smaller, more manageable sections. The size and shape of each cell are determined by the arrangement and spacing of these grids. By carefully designing the shape, size, and pointing of satellite beams, as well as the arrangement of grids, satellite communication systems can optimize coverage and communication efficiency. This is particularly important in the MLSCS.The available resources of SBSs. When determining the amount of bandwidth or data rate that SBSs can provide to each cell in a wireless communication network, it is important to consider not only the total available spectrum but also the spectrum usage policy in the area where the cell is located and the spectrum that is already occupied.The inter-beam interference. In the beam-hopping satellite communication system, the CCI among beams is a strong constraint. The position relationship between the SBS and the cell determines the azimuth and pitch of the beam pointing, and then the angle between any two beams can be derived from this, which can be used to measure the intensity of the CCI.The constraints of user fairness and other factors. Fairness is an important consideration in slot allocation among cells in a communication system. It refers to whether the allocation of slots is fair, ensuring that no cell is allocated too many slots while others are allocated too few. When slots are allocated unfairly, some cells may receive too many resources, while others may receive only a few resources or even none at all. This can lead to inefficiencies and unfairness in the system, as users in cells with fewer slots may not be able to access the resources they need to communicate effectively.

Given the problems and factors discussed above, SBSs should follow the following guiding principles when scheduling their beams:The cells should be within the service area. The scheduling of resources depends on whether the cells are located within the service area of the SBS.No CCI between any two beams. To avoid excessive CCI, the angle between any two working beams must comply with the interference threshold. Generally, a larger angle between beams results in lower levels of CCI.The fairness of slot allocation for all cells. When allocating slots, it is crucial to ensure that the ratio of allocated slots for each cell is proportional to its user demand or service load. However, it is equally important to avoid prioritizing cells with high service loads while neglecting those with lower loads. Ensuring that all cells receive an adequate level of service is necessary for overall network performance.

In addition, when designing and calculating beam-hopping patterns, three main spectrum usage schemes are considered. These schemes play a crucial role in optimizing network performance and resource allocation.

Full scheme. All beams use the full frequency band available to the SBS, so they must be separated by a certain angle to avoid and reduce inter-beam interference.Reuse scheme. The available spectrum of the SBS is divided into multiple non-overlapping sub-bands. To avoid interference between adjacent cells, frequency multiplexing is used, which is similar to multicolor multiplexing. This allows different sub-bands to be used simultaneously among adjacent cells without causing interference.Hybrid scheme. Some beams use the full frequency band, while others use only a partial frequency band. During beam scheduling, cells with overlapping frequency bands need to calculate the angle between their beams. If interference is detected, the beams cannot illuminate the cells at the same time slot.

The beam scheduling algorithm includes several sub-algorithms. The first is the azimuth and pitch calculation algorithm, which is used for beam pointing. This algorithm converts the latitude, longitude, and altitude of the SBS and the grid into 3D coordinates in the ECEF coordinate system. Then, it calculates the vector from the SBS to the grid to determine the azimuth and pitch of the SBS. The second sub-algorithm calculates the angle between any two beam pointings using the azimuth and pitch of each beam. This angle is used to determine whether interference exists between the beams. In conclusion, the beam scheduling algorithm, which includes these sub-algorithms, is shown in Algorithm 1.

In addition, to guarantee service in areas with sparse users and traffic, in addition to the service beam scheduling shown in Algorithm 1, the SBSs will perform a round-robin schedule of the signaling beams, periodically collecting data to determine if there are any users or traffic arrivals in all cells within their service areas. If so, the service beams will be scheduled through Algorithm 1 to provide service to all users.
**Algorithm 1:** Beam Scheduling Algorithm.**Require:** The position $P_{s}=\left\{l_{s},b_{s},h_{s}\right\}$ (longtitude, latitude, altitude) of SBS, the position $P_{u}=\left\{l_{u},b_{u},h_{u}\right\}$ of user/grid, user/service distribution $D=\{d_{1},d_{2},\ldots ,d_{M}\}$, the number of cells is *M*, available frequency band $B_{s}$ of SBS, the maximum number of available beams *N*, the slot set T.**Ensure:** The slot and beam assignment for cells.1:Check the table to match the grid with SBS’s footprint and the grids with each cell in the service area.2:Calculate the number of users or service volume of each cell in the SBS’s service area and generate the service volume set $C_{s}=\left\{c_{1},c_{2},\ldots ,c_{M}\right\}$ of cells.3:Calculate the proportion of service volume in each cell to the total service volume in the satellite service area.4:**for **i∈T**do**5:   $C_{s\_temp}=C_{s}$;6:   **for** j=1:N **do**7:       **while** the slots in $C_{s\_temp}$ larger than 0 **do**8:          Select the cell $c_{max}$ with the largest service volume;9:          **if** The slots assigned to $c_{max}$ is not larger than the proportion and the left service volume within it is greater than 0 **then**10:            **if** The CCI between $c_{max}$ and the other cells that have been assigned beams in slot *i* is less than interference threshold. **then**11:               Assign slot *i* and beam *j* to cell $c_{max}$.12:               Delete $c_{max}$ from $C_{s\_temp}$ or update its service volume with 0.13:               Update the left service volume of cell $c_{max}$ in $C_{s}$.14:               Break.15:            **else**16:               Update the service volume of cell $c_{max}$ in $C_{s\_temp}$ with 0.17:            **end if**18:        **else**19:            Delete $c_{max}$ from $C_{s\_temp}$ or update its service volume with 0.20:        **end if**21:     **end while**22:   **end for**23:**end for**

## 7. Performance Assessments

### 7.1. Assessment Metrics

Based on system design and resource scheduling, we evaluate the effectiveness of service area planning and resource scheduling. This evaluation aims to provide insights and recommendations to users, such as system administrators and stakeholders, for optimizing the system’s design. The evaluation focuses on the following four key aspects.

#### 7.1.1. Effectiveness Evaluation of Service Area

Maximum service area (MSA): This refers to the largest area that a satellite can cover, primarily determined by its orbit height. This area is relatively fixed and serves as the upper limit for the satellite’s coverage.

Actual service area (ASA): After planning the service area, the specific area served by each SBS can be determined.

This area can be calculated using a grid-based approach and represents the subset of the maximum service area that is actually covered by the SBS.

#### 7.1.2. Resource Utilization

Resource demands (RD): Calculate the actual bandwidth demands of each SBS based on real-time service demands.

Available resources (AR): Determine the system’s available spectrum resources allocated to each SBS, considering interference constraints that limit the amount of spectrum that can be used in a given area.

Resources actually used (RAU): Calculate the volume of resources (e.g., bandwidth) actually used by each SBS based on real-time service demands and the resources allocated to users.

Overall system resource usage (OSRU): Gather statistics on the usage of system resources across all SBSs to provide insights into resource utilization and potential areas for optimization.

#### 7.1.3. Number of Served Users

The number of users in the service area of SBS: This total count includes all individuals who fall within the SBS’s coverage area, regardless of their activity status.

The number of active users that need services in the SBS’s service area: Active users refer to those who have engaged with the SBS services within a specified time frame (e.g., the past month). Knowing this number helps in planning for current service demands and resource allocation.

The number of inactive users in the SBS’s service area: Inactive users are those who have not utilized the SBS services for an extended period (e.g., more than three months). Understanding the proportion of inactive users can indicate potential churn and highlight areas for re-engagement strategies.

#### 7.1.4. User Satisfaction

Service demand volume: This refers to the total user requests for service within the system or SBS’s service area over a specified period (e.g., monthly). It indicates the level of demand placed on the system.

Actual service volume (or throughput): This represents the amount of service successfully provided by the system or SBS during the same period. It reflects the system’s capacity to meet demand.

Degree of satisfaction: This metric indicates how well the system or SBS meets user demand, calculated as the ratio of actual service volume to service demand volume. A high degree of satisfaction (close to 1 or 100%) suggests good performance, while a lower degree might indicate underservicing or inefficiencies.

The complexity of satellite communication systems necessitates the development of a multi-level evaluation indicator system that incorporates both qualitative and quantitative indicators. The characteristics of this system are shaped by the interplay of various factors, reflecting the intricate nature of satellite communications. Generally speaking, the degree of system completion depends on the support of different capabilities and must fully reflect the overall needs of the system. To establish an effective evaluation system, it is crucial to adhere to the principles of systematicness, hierarchy, and adaptability. Systematicness ensures that all relevant aspects of the system are considered, hierarchy organizes the evaluation indicators into manageable levels, and adaptability allows the system to evolve with changes in technology and requirements. The hierarchical structure of the evaluation indicators, as shown in Table 2, provides a framework for organizing and evaluating the system. This structure is designed to capture the complexity of satellite communication systems in a structured and systematic manner. A brief elaboration of the hierarchical structure is as follows:Coverage. The coverage indicator is a critical aspect of SBSs, reflecting their coverage characteristics and their influence on system performance. It is primarily described by three key metrics: the single-satellite coverage ratio, the double-satellite coverage ratio, and the multi-satellite coverage ratio. The single-satellite coverage ratio represents the fraction of the Earth’s surface covered by only one satellite within the system. Users in this area can only be served by one satellite, resulting in limited robustness and flexibility. If the single satellite or its space–ground link fails, users in this area may experience service disruptions. In contrast, users in the double-satellite coverage area can be served by two SBSs. This dual coverage allows for seamless handover to another satellite when one space–ground link goes wrong, greatly improving robustness and scheduling flexibility. Users in the multi-satellite coverage area enjoy even higher levels of robustness and flexibility, as they can be served by multiple SBSs. However, this increased coverage comes with a trade-off: more complex interference patterns among the satellites, which can challenge system performance. To address these interference issues, practical and effective interference management strategies and methods are crucial. Techniques such as power control, which adjusts the transmit power of satellites to minimize interference, beamforming, which directs signals to specific areas to reduce unwanted interference, and frequency reuse planning, which optimizes the use of frequencies to avoid interference, are all essential tools in ensuring optimal system performance. By implementing these strategies, system designers can harness the benefits of multi-satellite coverage while mitigating the associated interference challenges.Capacity. Capacity is a critical indicator for measuring the performance of communication systems. A larger capacity allows for a higher information rate to be provided to users, directly enhancing the QoS. In the MLSCS, various capacity evaluation indicators are used, including single-channel capacity, single-beam capacity, single-satellite capacity, system (or network) capacity, and congestion probability. Single-channel capacity determines the maximum information rate that an SBS can provide to a single user. Single-beam capacity represents the capacity that a single beam of a satellite can provide to its coverage area, while single-satellite capacity defines the total capacity a single satellite can offer. System capacity, on the other hand, indicates the maximum capacity that the MLSCS can provide to users worldwide. Finally, congestion probability measures the likelihood of link congestion occurring under certain network load conditions, reflecting the system’s optimization capabilities in resource scheduling and load balancing.Multiple access. Multiple access directly reflects the system’s ability to provide services to users, which is determined by access resources, the air interface waveform, and the multiple access protocol. The access capacity represents the maximum number of users or transmission rate that can access simultaneously through a single channel, single beam, single satellite, or the entire system. The access delay is the duration from the moment a user sends an access request (in grant-based access) or data (in grant-free access) to the successful access. Grant-based access requires a user to request permission before transmitting, while grant-free access allows users to transmit data without prior permission. The access success probability is a crucial indicator, determined by the number of users attempting to access simultaneously, the multiple access protocol, available access resources, and multi-user detection (MUD) approaches.Quality of Service. QoS refers to the ability of a network to utilize various technologies to provide improved service to specified users and address issues such as network delay and congestion. QoS assurance is crucial for networks with limited capacity and resources, especially in applications that require a fixed transmission rate and are sensitive to delay, such as streaming multimedia and IoV applications. Therefore, similar to terrestrial networks, common QoS indicators such as the bit error ratio (BER), transmission rate, end-to-end delay, and call drop ratio are also included in the performance evaluation of the MLSCS.Adaptability. Adaptability reflects how well a system performs and includes terminal availability, reliability, scalability, maintainability, and networking flexibility. Terminal availability is measured based on the usability, learnability, human–machine interaction, response speed, stability, and security of user terminals. Reliability indicates the ability of a system, such as an SBS, to perform a specified mission without failure within a certain period and under certain conditions. In the MLSCS, reliability is measured by the degree of connectivity, redundancy, and connectivity probability. The degree of connectivity and connectivity probability determines the likelihood that service data have an end-to-end path in the system. Redundancy reflects the ability or probability of finding an alternative device or link when the current one fails. Scalability, maintainability, and networking flexibility depict the ability of the MLSCS to expand or add new entities, enrich functions, improve performance, recover from failures, and evenly change the network topology and service flows to form various subnets or specialized networks.

### 7.2. Simulation Results

To validate the performance improvement in the presented hierarchical resource management architecture for the MLSCS, we conducted several simulations using STK10 and Matlab 2018b in this section and the simulation parameters in Table 3.

Figure 10 presents the simulation results of the bandwidth utilization rate for both the two-stage game management and the single SBS management. The figure shows that the bandwidth utilization rate of the two-stage game management is approximately 37∼50% higher than that of the single SBS management. This significant improvement in bandwidth utilization results in substantial savings of spectrum resources in systems with limited spectrum availability.

The throughput per beam against the number of UTs in a cell is simulated and presented in Figure 11. As shown in the figure, with the increase in the number of users per beam, the throughput per beam gradually increases. However, due to the limited beam capacity, the increase in user numbers leads to collision issues, resulting in a non-linear increase that gradually plateaus. When user scheduling is implemented, compared to without user scheduling, the throughput improves by approximately 25%. Additionally, the throughput of high-order MODCOD sets is larger than that of low-order ones.

Figure 12 gives the throughput per SBS against the number of UTs in the coverage area of this SBS with user scheduling. The figure presents the simulation results of throughput under different modulation and coding schemes, comparing scenarios with and without beam scheduling (Algorithm 1) in a single SBS. As the number of users increases, the throughput gradually increases, but the growth is not linear and eventually reaches a bottleneck. In the case with beam scheduling, the throughput can be improved by over 30% at maximum, and the advantage becomes more pronounced as the number of users increases.

Figure 13 presents the system throughput with 288 SBSs against the number of UTs all over the world with user and beam scheduling. As can be seen from the figure, spectrum sensing and scheduling bring a certain gain to the system throughput. With the increase in the number of global users, the system throughput also increases, following a similar trend to that observed in Figure 12 and Figure 13. However, when spectrum sensing and scheduling capabilities are in place, the system throughput can be improved by up to 38% or more. Furthermore, the gain from spectrum resource scheduling becomes more significant as the number of users increases.

## 8. Conclusions and Future Works

### 8.1. Conclusions

This paper addresses the problems related to resource management and scheduling in the MLSCS, particularly focusing on the non-uniform distribution of users/services and resources, and the rapid changes in network topology caused by the dynamic characteristics of LEO satellites. These factors adversely affect system resource management and scheduling. To address these challenges, we propose a three-level resource management architecture, which includes an NCC, SBSs, and UTs. This architecture decouples resource management from quickly changing network conditions into three levels: global spectrum scheduling and service area planning at the NCC; time slot, beam, and channel resource management and access management at the SBSs; and user service data management and bandwidth requests at the UTs. Hierarchical resource management reduces management difficulty and improves system response speed, providing a guarantee for the resource scheduling of delay-sensitive services. In addition, we present the functions of the three-level management architecture. Based on this architecture, we study the influencing factors, scheduling approaches, and algorithms for global spectrum scheduling and service area planning. Finally, considering the characteristics of the MLSCS, we propose a performance evaluation system that includes several indicators, such as coverage characteristics, network capacity, access control, QoS assurance, and adaptability, and perform several simulations to validate the effectiveness of hierarchical resource management in the MLSCS.

### 8.2. Potential Limitations and Challenges

Although the hierarchical resource management architecture proposed in this paper is designed based on future actual systems, endowing the system with strong capabilities in global spectrum and service area planning, as well as satellite and user resource scheduling, there may be the following limitations and challenges to its practical application:Collection of spectrum data. The collection of global spectrum data requires the deployment of spectrum detection and monitoring equipment worldwide. Deploying such systems on land is obviously costly and impractical. A feasible approach is to equip satellite systems with spectrum detection and monitoring payloads. However, the satellite platform itself has a limited size and weight, and adding payloads requires a more integrated design. At the same time, the collected spectrum data need to be processed and transmitted in real time to support global spectrum planning at the NCC, which places higher demands on the satellite’s computing and processing capabilities, as well as the data transmission capacity of the entire network.Overhead of network status information. Service area planning is based on the distribution of users, services, and resources in various regions across the entire network. However, utilizing satellite-to-ground links inherently increases the consumption of satellite resources and payload burden. In cases where global station deployment is not feasible, this status information also needs to be transmitted to the NCC via inter-satellite links (ISLs), significantly increasing the processing load and overhead for satellites.Real-time resource scheduling. In the MLSCS, LEO satellites move rapidly, causing the users and areas they cover to change quickly and users to switch rapidly between different satellites and beams. Meanwhile, sudden interferences, changes in payload status, and variations in platform status can all cause abrupt changes in the network state. However, whether the update of status and the scheduling of resources can keep up with the changes in network status poses a significant challenge to the real-time design of the system’s resource management architecture and mechanisms.The application of novel technologies. The application of novel technologies, such as beamforming, beam-hopping, non-orthogonal multiple access (NOMA), and laser communication, has added new dimensions to resource scheduling in the MLSCS. The scheduling of multi-dimensional resources introduces additional constraints, placing higher demands on the effectiveness, reliability, and timeliness of resource scheduling algorithms.

### 8.3. Future Works

The mega-LEO satellite constellation represents a vast space communication system characterized by numerous nodes, a complex configuration, and a dynamic topology. Operating within a limited resource environment and an open electromagnetic setting, this system may lead to unpredictable outcomes. Consequently, resource scheduling becomes significantly more complex. In the future, several key considerations will need to be addressed to manage these challenges effectively:The global spectrum database. To create a comprehensive database of spectrum resource data, researchers must collect and collate information from various regions and satellite systems worldwide. This involves understanding international regulations set by the ITU, national and regional policies enforced by spectrum authorities, and the actual spectrum currently in use. The database should incorporate real-time updating and querying functions to facilitate efficient management and planning of spectrum resources.Endogenous intelligence in resource management. Adding artificial intelligence (AI) models or machine learning frameworks to big data processing platforms is a promising approach for analyzing resource scheduling data and recognizing patterns. To utilize these advanced tools effectively, researchers must first become familiar with the basic operations of AI frameworks. They also need to learn how to apply these tools to large datasets, ensuring they can extract valuable insights and make informed decisions.Fusion of AI and computing. To build an intelligent communication ecosystem in the industry, a distributed intelligent platform in the form of software is required. This platform provides intelligent services by supporting the hardware computing resources necessary for intelligent applications to run. Additionally, it offers connection services and integrates intelligence, computing, and connection through network resources. However, this increases the complexity of managing and scheduling integrated services. To achieve optimal scheduling and arrangement, comprehensive service discovery, process arrangement, resource scheduling, and performance measurement are necessary.Novel beam-hopping strategy. Beam-hopping resource allocation with a full frequency band is a novel direction that considers priority, service data scheduling, and network load balance. In this study, these factors are treated as critical inputs for resource allocation algorithms. To account for these inputs, algorithms may need to be modified in several ways. For example, different priorities can be assigned to various types of services or users to ensure that high-priority services have sufficient bandwidth and time slots when resources are limited. Ensuring efficient service data scheduling and maintaining load balance across the network are also important considerations.

## Figures and Tables

**Figure 1 sensors-25-00902-f001:**
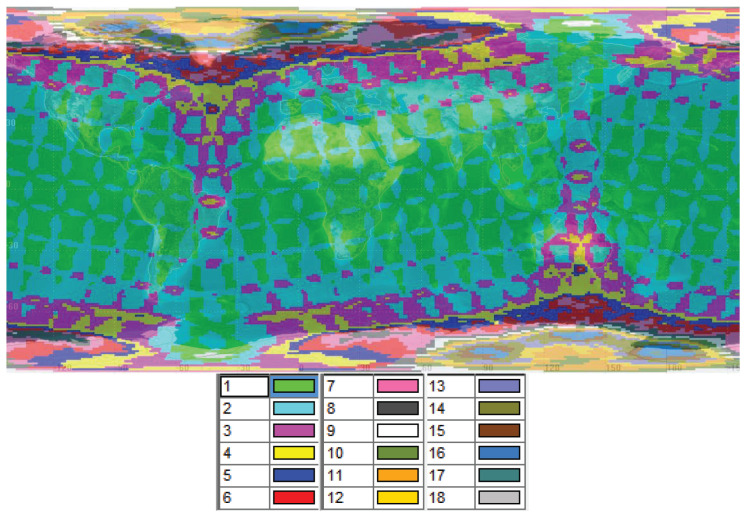
The global coverage of a Walker LEO constellation.

**Figure 2 sensors-25-00902-f002:**
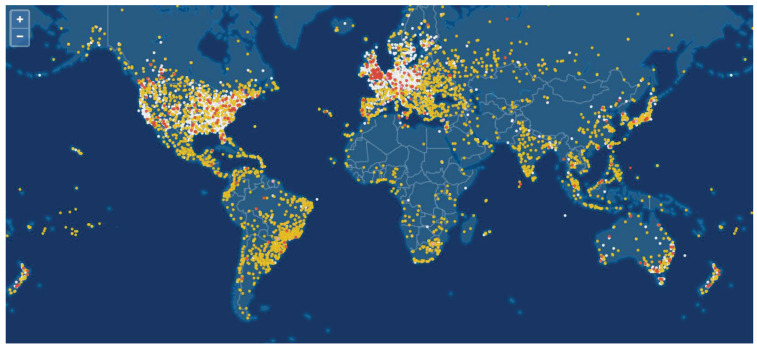
The user distribution of Iridium system. Red dots represent high-density areas, yellow dots represent medium-density areas, white dots represent low-density areas, and areas with no dots indicate no users in that area.

**Figure 3 sensors-25-00902-f003:**
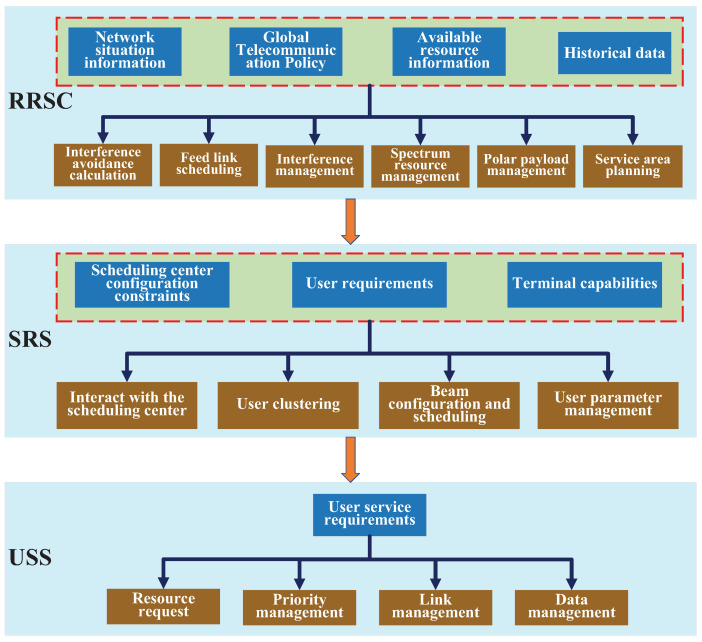
Hierarchical resource management architecture of MLSCS.

**Figure 4 sensors-25-00902-f004:**
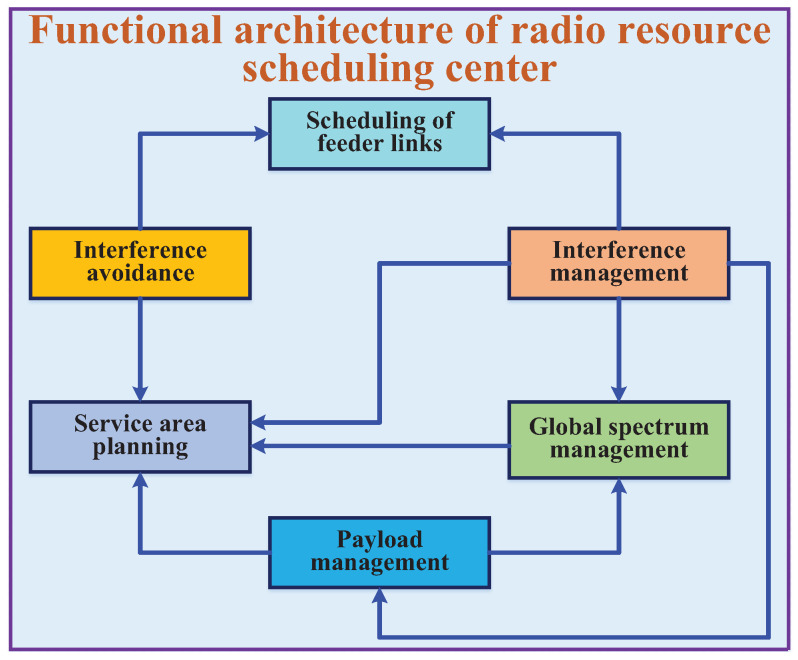
Functional architecture of radio resource scheduling center.

**Figure 5 sensors-25-00902-f005:**
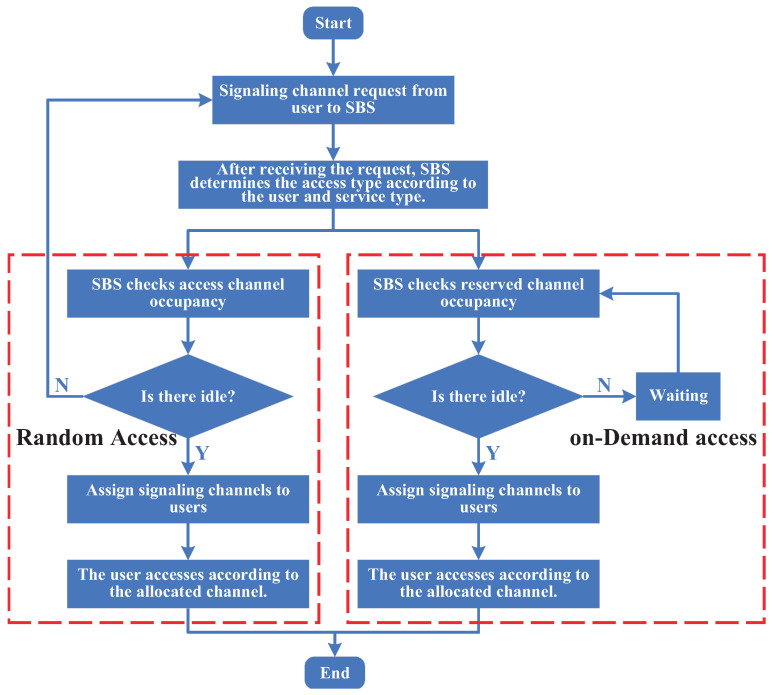
Signaling channel access procedure.

**Figure 6 sensors-25-00902-f006:**
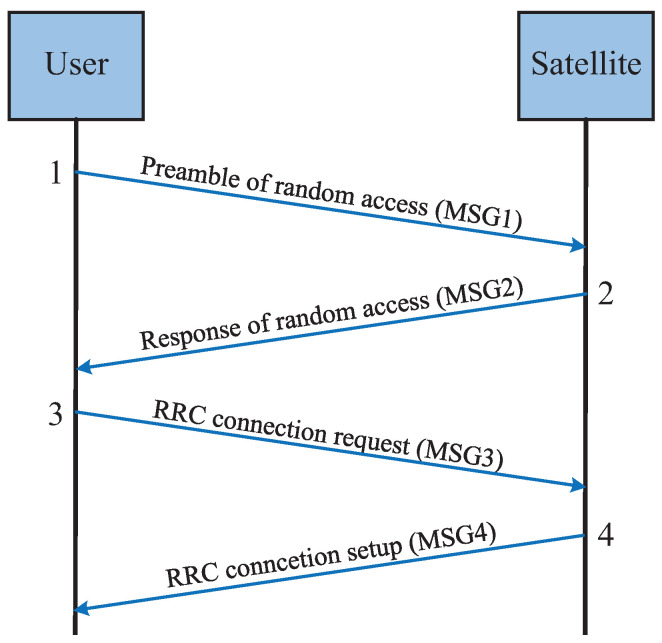
Grant-based random access procedure.

**Figure 7 sensors-25-00902-f007:**
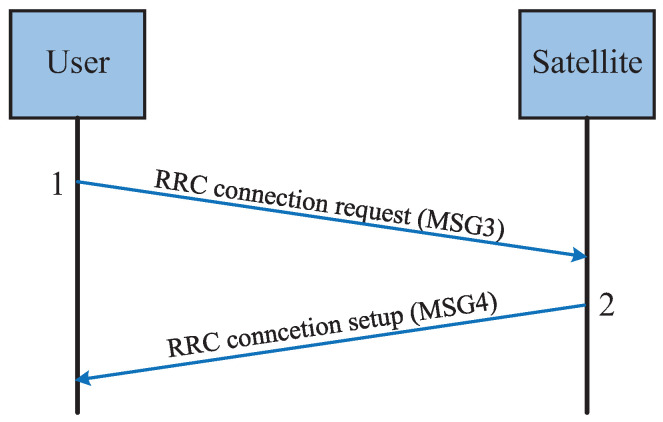
Grant-free random access procedure.

**Figure 8 sensors-25-00902-f008:**
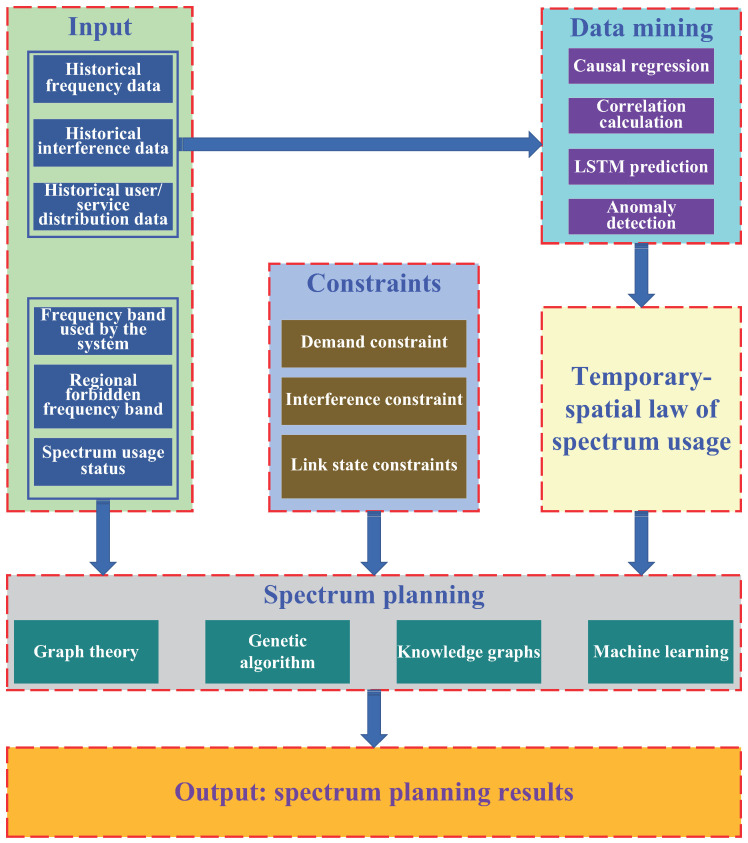
Function modules of spectrum planning.

**Figure 9 sensors-25-00902-f009:**
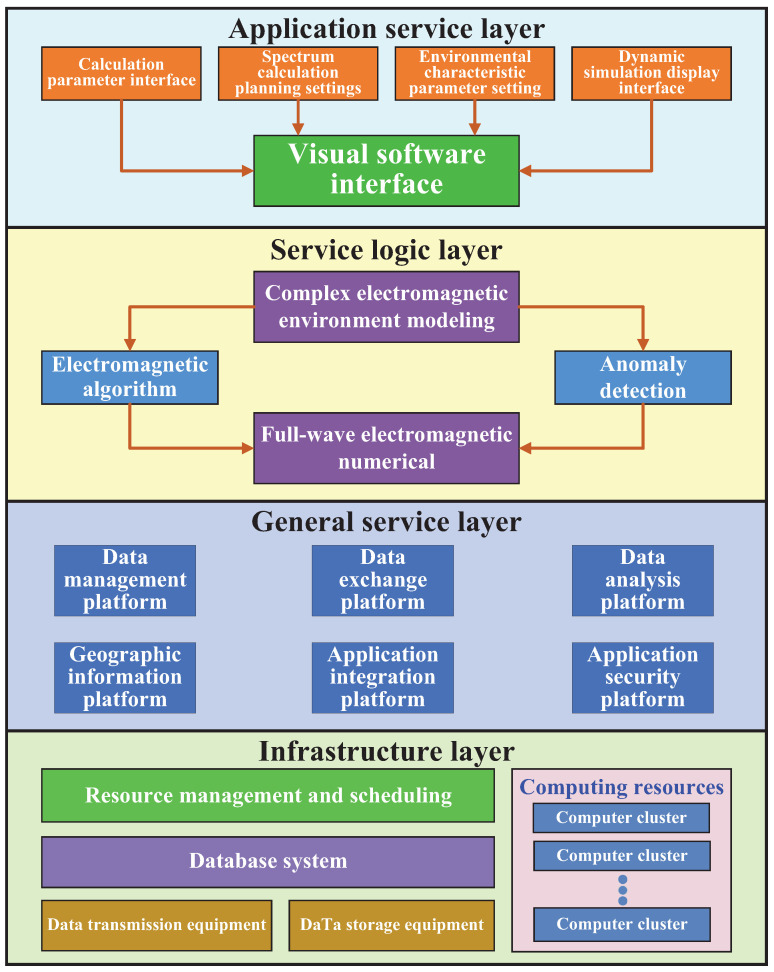
Software architecture of spectrum planning module.

**Figure 10 sensors-25-00902-f010:**
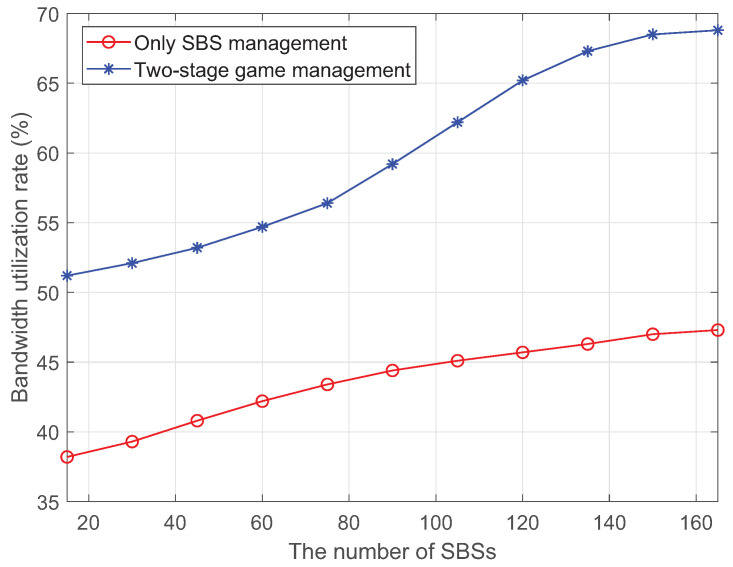
Bandwidth utilization rate of only SBS management and two-stage game management.

**Figure 11 sensors-25-00902-f011:**
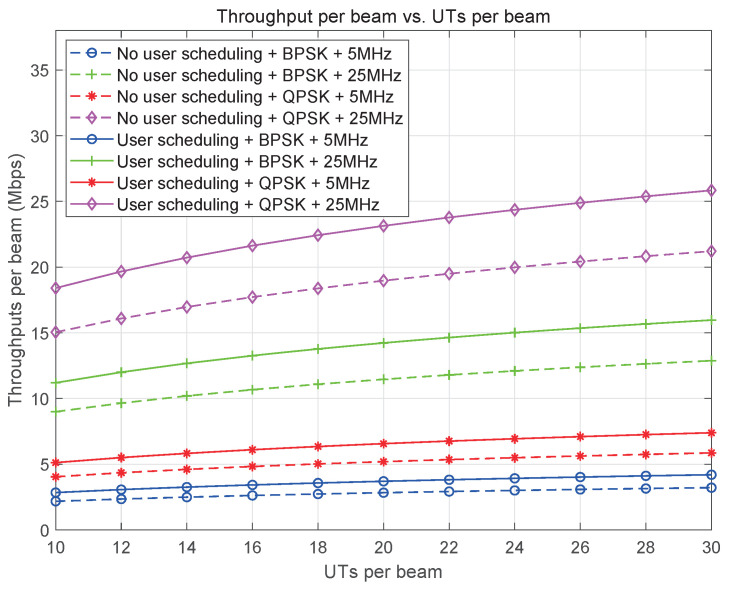
Throughput per beam (Mbps) with different number of UTs and different MODCOD sets.

**Figure 12 sensors-25-00902-f012:**
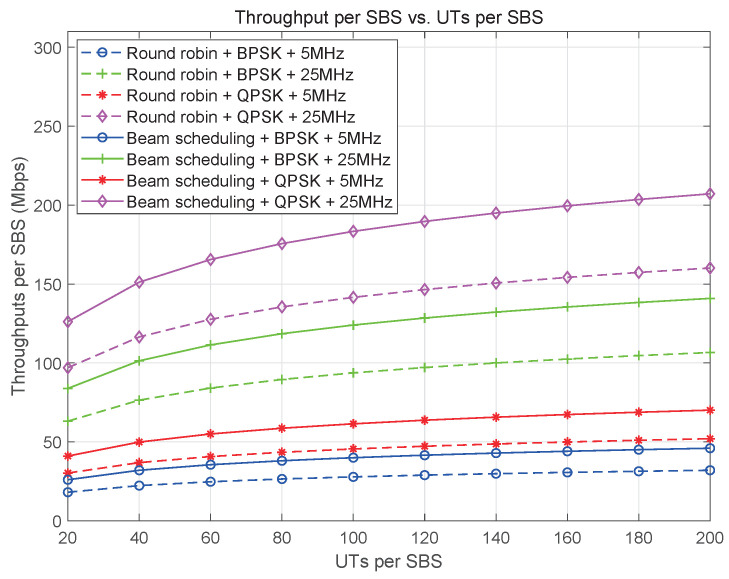
Throughput per SBS (Mbps) vs. UTs per SBS with different beam scheduling schemes and MODCOD sets.

**Figure 13 sensors-25-00902-f013:**
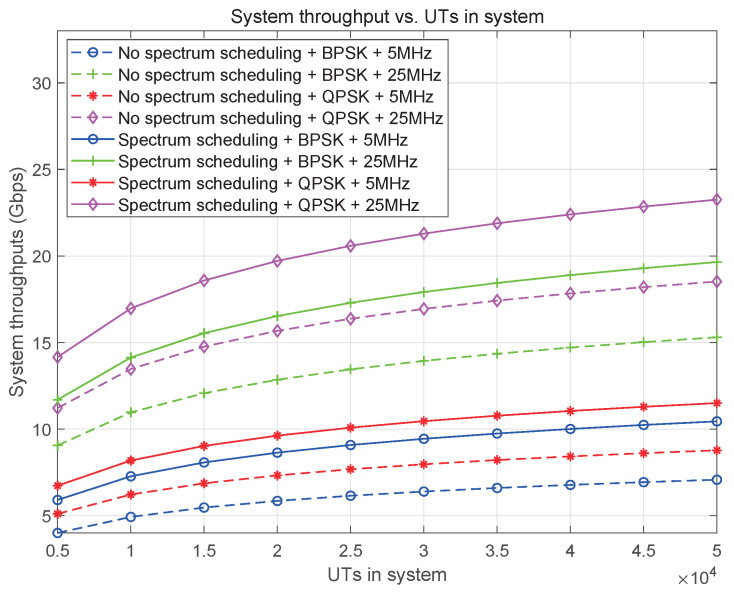
System throughput (Gbps) vs. UTs in system with different spectrum scheduling schemes and MODCOD sets.

**Table 1 sensors-25-00902-t001:** Summary of state-of-the-art approaches for resource allocation in terms of performed tasks, approaches, drawbacks, and applications.

Reference	Task	Approach	Drawback	Application
[5] (2020)	Resource allocation	Lagrange dual method	The approach relies on high-altitude platforms (HAPs) and terrestrial relays (TRs)	Simulated environments
[8] (2022)	Resource allocation	EMCL	Not LEO constellation	Simulated environments
[13] (2022)	Resource allocation	MDP and DRL	Only uplink transmission in LEO networks with multiband antennas	Simulated environments
[15] (2022)	Interference control	Online frequency allocation	Downlink spectrum sharing in a two-satellite system	Simulated environments
[17,18,19,20,21] (2018–2022)	3C resource management	Tapped water-filling algorithm, Nash bargaining game and resource map	The optimization target is throughput without considering other factors	Simulated environments
[24,25,26,27,28,29] (2019–2022)	Resource allocation	Q-learning, DRL, and CR	The emphasis is on joint allocation in TSNs	Simulated environments
[30,31,32] (2019–2022)	Resource allocation	Contract theory, artificial bee colony algorithm, and iterative algorithm	No consideration of mega-LEO constellation and the optimization target is single	Simulated environments
[33,34,35,36] (2021–2024)	Resource allocation	Precoding algorithm and MWC	The focus is on beam-hopping scheduling	Simulated environments
[37] (2021)	Resource allocation	Traffic prediction (TP) and resource matching (RM)	Only consider the regional scenario	Simulated environments
[38] (2023)	Resource allocation	Lyapunov optimization and successive convex approximation	Focus on satellite–air communications	Simulated environments
[39] (2023)	Resource allocation	Distributed algorithm	Only considers the downlink	Simulated environments
[40] (2023)	Resource allocation	Multi-agent multi-armed bandit	Two-way transmission in satellite communication systems may be infeasible sometimes	Simulated environments
[41] (2023)	Resource allocation	RL	Focus on mobility management	Simulated environments
[42] (2024)	Resource allocation	Breadth-first search (BFS)	Focus on edge computation and computation offloading	Simulated environments
[43] (2023)	Resource allocation	ML	Focus on artificial intelligence for satellite communication	Analysis
[44] (2024)	Network control	Design	Focus on the control of LEO giant constellations	Simulated environments

**Table 2 sensors-25-00902-t002:** Assessment indicators.

The First Level	The Second Level	The Third Level
Coverage	Coverage rate	Coverage rate of single satellite
		Coverage rate of double satellites
		Coverage rate of multiple satellites
	Outage performance	Maximum outage time
		Average outage time
		Outage probability
	Elevation	Minimum elevation
		Average elevation
Capacity	Capacity of single channel	
	Capacity of single beam	
	Capacity of single satellite	
	System capacity	
	Congestion probability	
Access and control	Access capacity	
	Access delay	
	Access success probability	
Quality of service	Bit error rate	
	Transmission rate	
	End-to-end delay	
	Call drop rate	
Adaptability	Terminal availability	Usability
		Learnability
		Human–machine interaction
		Response speed
		Stability
		Security
	Reliability	Degree of connectivity
		Redundancy
		Connectivity probability
	Scalability	
	Maintainability	
	Networking flexibility	

**Table 3 sensors-25-00902-t003:** Simulation parameters.

Parameters	Value
Constellation type	Polar walker
Number of satellites	288
Number of beams per satellite	8
Altitude of LEO satellites	1025 km
Speed of satellites	7.447 km/s
LEO satellite 3 dB beamwidth	3°
LEO satellite coverage radius	1189 km
LEO beam coverage radius	53.8 km
Number of user terminals	10,000
User distribution	Random
Mean value of Poisson arrival for service	0.3
Interval of beam-hopping	200 ms
Cycle of beam-hopping	10 s
Maximum total power of LEO satellites	1000 W
Maximum power on each subchannel of LEO satellites	40 W
LEO satellite antenna η	20 dB
Terminal antenna η	20 dB
Noise power spectral density	−173 dBm/Hz
Average radius of hot-spot regions	10 km
Carrier frequency of Ka band	30 GHz
Bandwidth	5/25 MHz
Coding	$\frac{1}{2}$Turbo
Modulation	BPSK/QPSK
Packet size	1024 bits

## Data Availability

Data are contained within the article.

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
