# Peer review of "Hierarchical Resource Management for Mega-LEO Satellite Constellation"

_sensors, 2025, doi:10.3390/s25030902_

Round 1
Reviewer 1 Report
Comments and Suggestions for Authors
The authors propose an interesting and well-developed work, but its presentation is somewhat chaotic due to the abundance of information. Improving the organization and clarity of the presentation would enhance its comprehensibility. Here are some recommendations for improvement:
- Including a comparative table of related works would be beneficial, especially since this paper has a survey-like nature and there are numerous surveys on this topic.
- A clear description of the methodology is missing and should be included.
- Adding a diagram at the beginning of the paper to outline its structure and contents would provide readers with a clear overview, especially given the extensive nature of the content.
- The meaning of the colored numbers is neither explained in the figure 1 legend nor in the text. This should be clarified.
- Figure 2: A color legend is needed to improve its interpretability.
- Some section beginnings and paragraphs start with lowercase letters, which need correction.
- While the title emphasizes LEO constellations as the main topic, some sections discuss UAVs. The integration of UAVs with LEO constellations needs to be clarified.
- Certain sections appear unnecessary for explaining just a few lines of text. These could be condensed or integrated into other sections.
- A diagram or figure, similar to Figure 3, could also illustrate the three levels of architecture.
- Too few references exist for a widely studied topic like resource management. A more extensive citation of relevant works would strengthen the context.
Additionally, the absence of ML techniques for this type of task is noteworthy. The following survey includes a dedicated section on using ML and DL techniques for radio resource management in megaconstellations.
[Ref] Fontanesi, G., et al (2023). Artificial Intelligence for Satellite Communication and Non-Terrestrial Networks: A Survey. ArXiv. https://arxiv.org/abs/2304.13008
Why are such techniques not necessary for the proposed architecture? What advantages does the proposed approach have over these techniques? This comparison should be discussed in the introduction, background, or methodology section better to highlight the authors' contributions to this paper.
Reviewer 2 Report
Comments and Suggestions for Authors
This paper investigates hierarchical resource management for the mega LEO satellite constellation. The topic is interesting and valuable. The theoretical derivation is rigorous, and the results appear correct and believable. The manuscript is generally well-written (although I provided many corrections and suggestions) and provides an interesting conclusion. Nevertheless, the reviewer has some concerns about this work, which you can find below. I suggest that the authors revise and improve the manuscript accordingly.
To effectively convey the paper's motivation, it is recommended to include a table that succinctly summarizes and compares it with existing literature.
The utilization of older references for the literature review limits the work's alignment with current research directions. Updating the references would provide better insight into the contemporary state of the field. The following related references should be reviewed to help the readers and prevent misleading of previously undertaken research works, doi: 10.1109/TITS.2023.3330419, 10.1109/TWC.2024.3452642, 10.1145/3636534.3649362, 10.1049/cmu2.12725.
How to obtain eq. (1). Please provide more explanations.
Which simulator is used to simulate? Also, include the parameters considered in a tabular form.
Minor:
8. conclusion Conclusion
Comments on the Quality of English Language
The English could be improved to more clearly express the research.
Reviewer 3 Report
Comments and Suggestions for Authors
Despite its strengths, the study has some notable weaknesses. First, many sections are lengthy and could be summarized more and more. The three-layer architecture is well-defined, the paper could benefit from a more thorough discussion on the potential limitations, challenges associated with implementing such a hierarchical structure in real-world scenarios. Optimization is questionable. For instance, issues related to latency, scalability, and the integration of existing infrastructure with new systems are not adequately addressed. Plus, while the simulations demonstrate improvements in resource management, there is a lack of detailed information regarding the parameters used in these simulations and how they reflect real-world conditions. This raises questions about the practical applicability of the results. Moreover, the paper could enhance its impact by including a comparative analysis with existing resource management frameworks or by discussing how the proposed methods stack up against alternative approaches. Lastly, while the focus on hotspot areas is pertinent, the implications for regions with lower user density are not explored, which could limit the generalizability of the findings.
Reviewer 4 Report
Comments and Suggestions for Authors
This paper is a solid work for LEO satellite constellation by exploring various resource scheduling strategies, approaches, and algorithms, including spectrum cognition, interference coordination, beam scheduling, multi-satellite collaboration, and random access. There are still some issues that need to be addressed.
1. Written quality should be improved. In 3.2.1. "due" should be "Due". In 3.2.4, "a mega" should be "A mega". In 3.2.5, "limited" should be "Limited".
2. Figures should be compiled in .eps or .pdf.
3. The authors should clarify the difference from previous survey paper.
4. The title of Section 8 is in wrong format.
Round 2
Reviewer 1 Report
Comments and Suggestions for Authors
Thanks for addressing my comments. I do not have any more questions.
Author Response
Dear reviewer,
Thanks very much for your work concerning our manuscript entitled “Hierarchical Resource Management for Mega LEO Satellite Constellation” (sensors-3371365). You have no more question. Based on the instructions provided in your letter, we uploaded the file of the revised manuscript.
Sincerely.
Gou Liang.

Reviewer 3 Report
Comments and Suggestions for Authors
The revisions are not satisfactory. Rejected.
Comments on the Quality of English LanguageGood.
Author Response
Dear reviewer,
Thanks very much for your work and reviewers’ comments concerning our manuscript entitled “Hierarchical Resource Management for Mega LEO Satellite Constellation” (sensors-3371365). These comments are valuable and very helpful. We have read through comments carefully, and have made corrections and interpretation. Based on the instructions provided in your letter, we uploaded the file of the revised manuscript. Revisions in the text are shown using blue highlight. The responses to your comments are marked in red and presented following.
We would like to thank you for allowing us to resubmit a revised copy of the manuscript and we highly appreciate your time and consideration.
Sincerely.
Gou Liang.
Q1. First, many sections are lengthy and could be summarized more and more. The three-layer architecture is well-defined, the paper could benefit from a more thorough discussion on the potential limitations, challenges associated with implementing such a hierarchical structure in real-world scenarios.
Response: Thank very much for your comments. We have taken your advice and added a section titled "Potential limitations and challenges" in the Conclusion of the article to elaborate on the limitations and challenges that the system may face. Meanwhile, we delete some paragraphs to make our paper more concise and thank you very much for this comment.
Q2. Optimization is questionable. For instance, issues related to latency, scalability, and the integration of existing infrastructure with new systems are not adequately addressed.
Response: Thank very much for your comments. One of the original purposes of designing a hierarchical architecture is to reduce signaling overhead and control delay, and to avoid the large overhead and long control loop caused by NCC centralized control. The limitation and challenges related to MLSCS is summarized in the conclusion of this paper which indicates that the overhead and delay to collect all network state information is notable and need to be considered carefully. However, the evaluation of the entire system overhead, delay and scalability needs to build a larger simulation verification system or platform, which is costly and time-consuming. But as a research direction, we will gradually implement and verify in the future work.
Q3. Plus, while the simulations demonstrate improvements in resource management, there is a lack of detailed information regarding the parameters used in these simulations and how they reflect real-world conditions. This raises questions about the practical applicability of the results.
Response: Thanks for your suggestions. Some parameters have been added into Table 3 and some outcomes are difficult to verify under the existing conditions. These parameters include LEO constellation parameters, user distribution and traffic parameters, as well as satellite communication parameters, which can be reflect the real-world conditions from network to node, and from service to links.
Q4. Moreover, the paper could enhance its impact by including a comparative analysis with existing resource management frameworks or by discussing how the proposed methods stack up against alternative approaches.
Response: Thank you very much for your advice. At present, some resource allocation and management architectures for LEO giant constellations have been proposed. However, most of them are optimized for specific scenarios or single indicators. There is no outcome for the overall consideration of resource management for mega LEO constellation. And our paper is the first effort to build a relatively complete resource management system for MLSCS. Hence, it is difficult to find objects for comparative analysis of the overall architecture at present and we are very sorry for it.
Q5. Lastly, while the focus on hotspot areas is pertinent, the implications for regions with lower user density are not explored, which could limit the generalizability of the findings.
Response: Thank you very much. In the process of service area scheduling, global coverage and service area planning are taken into account, even in areas with sparse users and traffic, satellite resources are allocated. Simultaneously, through round-robin of the signaling beams, user and service information is obtained. Based on this, service beams are scheduled to ensure service provision to all users. This mechanism is supplemented with additional explanations at the end of sections 6.3 and 6.4.
Thanks for your review very much again. Wish you all the best.

Round 3
Reviewer 3 Report
Comments and Suggestions for Authors
Although pages are lengthy but it is acceptable.